# Rising from Ashes: Generalized Federated Learning via Dynamic Parameter Reset

**Jiahao Wu[1], Ming Hu[2*], Yanxin Yang[1], Xiaofei Xie[2], Zekai Chen[1],**
**Chenyu Song[1], Mingsong Chen[1*]**
[1]MoE Eng. Research Center of SW/HW Co-Design Tech. and App., East China Normal University
[2]School of Computing and Information Systems, Singapore Management University
hu.ming.work@gmail.com, mschen@sei.ecnu.edu.cn

## Abstract

Although Federated Learning (FL) is promising for privacy-preserving collaborative model training, it suffers from low inference performance due to heterogeneous client data. Due to heterogeneous data across clients, FL training easily learns client-specific overfitting features. Existing FL methods adopt coarse-grained averaging, which can easily cause the global model to get stuck in local optima, leading to poor generalization. Specifically, this paper presents a novel FL framework, FedPhoenix, to address this issue. It stochastically resets partial parameters in each round to destroy some features of the global model, guiding FL training to learn multiple generalized features for inference rather than specific overfitting features. Experimental results on various well-known datasets demonstrate that compared to SOTA FL methods, FedPhoenix can achieve up to 20.73% higher accuracy. The implementation is publicly available at **https://github.com/UniString/FedPhoenix**.

## 1 Introduction

With Artificial Intelligence (AI) technologies widely used in privacy-sensitive applications, protecting data privacy during model training has become an urgent requirement. To address this issue, Federated Learning [1–7] as a privacy-preserving distributed machine learning paradigm has been proposed and used in various AI applications, such as Artificial Intelligence of Things (AIoT) systems [8–12], recommender systems [13, 14], and healthcare systems [15, 16], which enables multiple clients to collaboratively train a global model without exposure of their raw data. Specifically, conventional FL consists of a cloud server and multiple clients. In each FL training round, the cloud server sends the global model to each client. Clients use their raw data to train the received model and then upload the trained model back to the cloud server. By aggregating all uploaded models, the cloud server can generate a new global model for the next round of training.

However, due to clients' diverse preferences and data volumes, client data is often not Independent and Identically Distributed (non-IID), leading to the "client drifting" problem [17–20]. Specifically, different data distributions lead to different optimization directions for local models. The traditional FedAvg strategy aggregates all local models to generate the global model, which may cause the global model's optimization directions to deviate from the optimal direction, leading it to easily get stuck in local optima. To optimize the FL training process, existing methods attempt to utilize correction terms [21, 17], knowledge distillation (KD) [22–25], model mutation [26, 27], client clustering [28–31], and multi-model searching [32–34] strategies.

---

[*]Ming Hu and Mingsong Chen are corresponding authors.

39th Conference on Neural Information Processing Systems (NeurIPS 2025).

Although these methods can improve the inference accuracy of the global model in non-IID scenarios, they introduce additional overhead or privacy leakage risks. For example, correction term-based methods require the communication overhead for the transmission of global correction terms. Knowledge distillation-based methods require additional public data or incur computation overhead to generate proxy datasets. Clustering-based and multi-model searching-based methods cannot support secure aggregation [35], which may increase the risk of privacy leakage. Therefore, *how to improve the performance of FL in non-IID scenarios without additional overheads or public data while integrating secure aggregation is a significant challenge in FL.*

Typically, compared to centralized model training, each client in an FL setting has access to far less data, which increases the risk of overfitting to local features. To improve the global model's inference performance, the local model should learn more general features rather than those specific to the local dataset. Unfortunately, due to limited access to local data, the cloud server cannot justify the quality of the global model. In addition, since each client cannot access other clients' data, it cannot determine which features are generalized. Intuitively, to avoid the global model overfitting to client-specific features, FL training should encourage the model to perform inference using more features rather than a few specific ones. Intuitively, if the model inference depends on multiple features, losing or destroying a few of them has only a minor impact on the final decision. In contrast, if the model relies on just a handful of features, destroying any one of them can severely affect its decisions.

Inspired by the above intuition, this paper presents a novel FL framework, *FedPhoenix*, to address the low training performance in FL. Like the Phoenix rising from the ashes, FedPhoenix randomly resets partial parameters of the global model to destroy certain features, generating different local models with different parameter resets for local training within a communication round. In this way, if a destroyed feature is unique to a particular device, retraining will be an alternative. In contrast, if the feature is truly generalized, it will be restored as FL training progresses. In addition, when features with large weights are destroyed, the model will try to use other features for inference while learning new features. At this time, the weights of some generalized features will increase, while those of device-specific features will decrease. To ensure the model's convergence efficiency, FedPhoenix includes a dynamic reset mechanism that adjusts the ratio of reset parameters as FL training progresses. The main contributions of this work are summarized as follows:

- We present a novel FL framework, FedPhoenix, that employs a heuristic parameter resetting mechanism to achieve generalized FL model training.
- We propose an adaptive parameter reset strategy to control the resetting ratio and range of each layer and a dynamic stabilization strategy to ensure the convergence of FedPhoenix.
- We present a theoretical analysis of FedPhoenix's convergence in convex settings and conduct comprehensive experiments on well-known benchmarks to demonstrate its effectiveness and convergence in non-convex settings.

## 2  Preliminary and Related Work

### 2.1  Problem Formulation

Consider a federated learning system with $N$ clients, each holding a local dataset $\mathcal{D}_i = \{z_j^{(i)}\}_{j=1}^{n_i}$, where $z_j^{(i)} = (x_j^{(i)}, y_j^{(i)})$ is a data sample. The goal is to collaboratively learn a global model $w \in \mathbb{R}^d$ by minimizing the objective:

$$\min_w F(w) = \sum\nolimits_{i=1}^{N} \frac{n_i}{n} F_i(w),$$

where $F_i(w) = \frac{1}{n_i} \sum_{j=1}^{n_i} \ell(w; z_j^{(i)})$ $n = \sum_{i=1}^{N} n_i$ is the total number of samples. Here, $\ell(\cdot)$ is a sample-wise loss (e.g., cross-entropy). The server coordinates iterative training: clients locally update the global model on their data, and the server aggregates these updates (e.g., via weighted averaging) to refine the global model.

### 2.2  Related Work

To address non-IID challenges in FL, existing methods often adopt various optimization strategies that come with inherent limitations. FedProx [21] introduces a regularization term to align local

models with the global model, reducing client drift by penalizing deviations. While this approach incurs minimal communication overhead, its accuracy improvement is often limited, especially in highly heterogeneous scenarios. ClusterSampling [28] groups clients by data or model similarity to improve aggregation representativeness. However, its computational complexity scales quadratically with the number of clients, making it impractical for large-scale deployments. FedGen [23] leverages knowledge distillation with synthetic proxy data to achieve knowledge transfer among clients. FedMut [26] injects diversity by mutating the global model through gradient perturbations. While this method demonstrates some effectiveness in exploring the solution space, its reliance on random mutations can lead to inconsistent updates, potentially destabilizing convergence. In addition, most existing methods inevitably incur additional communication overhead or require public data, or cannot be employed in secure aggregation, which significantly limits generalization.

To the best of our knowledge, FedPhoenix is the first FL method to use the heuristic parameter reset strategy to enhance generalization without additional data or communication overhead, and it is compatible with the secure aggregation strategy.

## 3 Prestudy and Our Motivation

**Intuition.** In non-IID scenarios, the limited diversity and volume of local data often cause the global model to overfit specific local features. Intuitively, the model inference process is more focused on a few overfitting features rather than on multiple generalized features, resulting in low inference performance for the global model trained by FedAvg.

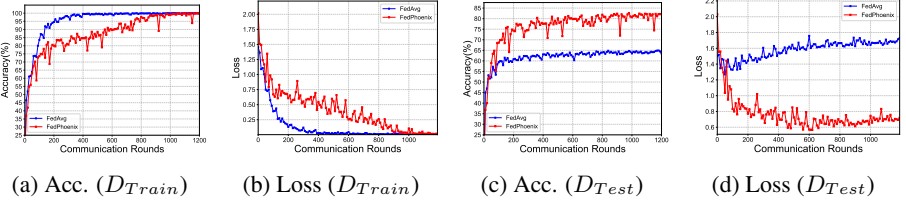

(a) Acc. ($D_{Train}$)  (b) Loss ($D_{Train}$)  (c) Acc. ($D_{Test}$)  (d) Loss ($D_{Test}$)

Figure 1: Accuracy and loss curves of FedAvg and FedPhoenix with $\alpha = 0.6$

**Study on the Overfitting Issue of the Global Model.** To explore the overfitting issue of FL, we conducted a prestudy for FedAvg on the CIFAR-10 dataset using the ResNet-18 model with the non-IID setting of the Dirichlet Distribution ($\alpha = 0.6$). Figure 1 presents accuracy and loss curves of FedAvg and our FedPhoenix on the training dataset and testing dataset. From Figure 1, we can observe that although the global model trained by FedAvg can eventually achieve 100% accuracy and close to 0 loss in the training dataset, its performance is still seriously limited in the testing dataset. We can find that the loss in the testing dataset initially decreases and then increases as training progresses. Based on the above observations, we can find that *FedAvg-based FL easily overfits to the training dataset.*

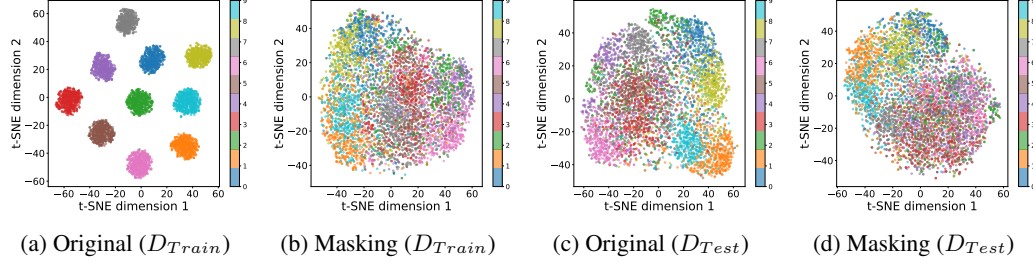

(a) Original ($D_{Train}$)  (b) Masking ($D_{Train}$)  (c) Original ($D_{Test}$)  (d) Masking ($D_{Test}$)

Figure 2: t-SNE for representations of the model trained by FedAvg

**Study on Learned Features of the Global Model.** To investigate whether the global model relies on a small number of overfitted features, we stochastically masked 10% of the parameters in a FedAvg-trained global model to remove certain features. We then collected intermediate representations from both the original and the masked models using the training and test datasets, categorizing them by label. Figure 2 illustrates the similarities between the representations of the original global model and the masked model on both the training and test datasets using t-SNE [36]. We observe that, in the training dataset, the intermediate representations of the original model across different

categories form distinct clusters. Nonetheless, when 10% of the parameters are masked, it becomes challenging to differentiate these clusters. In the test dataset, the representations of the original model across different categories overlap to some extent, indicating less distinct clustering. After masking 10% of the parameters, the representations of all categories in the test dataset become nearly indistinguishable. Therefore, masking a small number of features can cause significant serious performance degradation in the global model trained via FedAvg-based FL.

To further investigate the impact of a small subset of features on models trained by FedAvg, we evaluate the importance of each feature in a specific layer on the CIFAR-10 test dataset for each class by individually masking a single feature and measuring the accuracy degradation for that class. Then, we test the accuracy on the test dataset for the model trained by FedAvg with masking i) the top 16 important features (accounting for 3.125% of the total features) and ii) the stochastic 16 features of the last convolutional layer, respectively. As shown in Table 1, for models trained with FedAvg, masking the top 16 features results in significant accuracy degradation, while masking the stochastic 16 features has little effect on accuracy. Based on the above observations, we find that *the inference of the global model of FedAvg relies on only a few specific features rather than more generalized ones, making it vulnerable to parameter masking*.

**Our Idea.** Motivated by the aforementioned observations and results, this paper aims to guide the model to acquire more generalized features and to make inferences using a wide range of features rather than relying on just a few specific ones. To achieve this goal, we propose FedPhoenix, which randomly resets a small subset of the global models parameters to generate local models for FL training. In this way, the cloud server can destroy a few features of the global model. Since most clients can

Table 1: Impact of Partial Feature Layer Output Masking on Class-wise Accuracy

| Class | Accuracy of FedAvg (%) | | |
|---|---|---|---|
| | Original | Top16 | Random16 |
| Class 0 | 64.10% | 38.80% | 63.30% |
| Class 1 | 70.80% | 47.30% | 72.30% |
| Class 2 | 54.60% | 33.40% | 54.10% |
| Class 3 | 41.70% | 22.10% | 43.10% |
| Class 4 | 64.40% | 39.40% | 64.50% |

learn generalized features, these features can be quickly restored. On the contrary, the specific local features unique to a few clients are hard to restore. In addition, since features with large weights are randomly reset, parameter resetting can guide the model to adjust its weights to learn more features for inference rather than just a few.

To validate the effectiveness of our idea, we compare the performance of FedPhoenix with that of FedAvg. As shown in Figure 1, FedPhoenix can greatly alleviate overfitting and achieve higher inference accuracy on the test dataset.

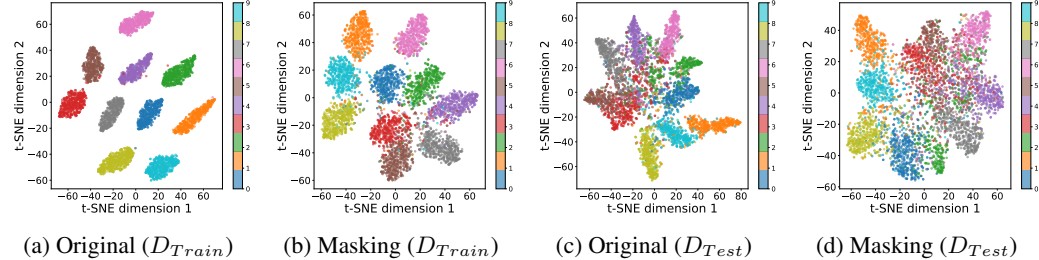

(a) Original ($D_{Train}$)    (b) Masking ($D_{Train}$)    (c) Original ($D_{Test}$)    (d) Masking ($D_{Test}$)

Figure 3: t-SNE for representations of the model trained by FedPhoenix

Figure 3 visualizes the similarities of representations of the original global model and the masked model trained by FedPhoenix on the training dataset and test dataset using t-SNE. We observe that representations of the original and the masked models across different categories form distinctly separate clusters in both the training and test datasets, indicating that the global model trained by FedPhoenix learns more generalized features and utilizes more features for inference than the model trained by FedAvg.

## 4 Our Phoenix Approach

### 4.1 Overview

Figure 4 illustrates the framework and workflow of our FedPhoenix approach, which includes a central cloud server and multiple local clients. In each FL training round, the cloud server selects $K$

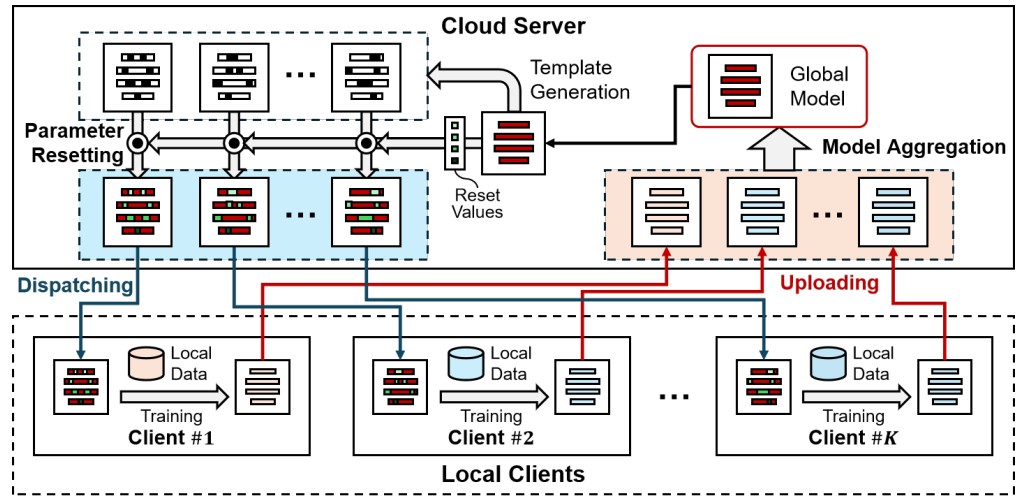

Figure 4: Framework and workflow of FedPhoenix

activated clients to participate in local training. Assuming that there are $N$ clients, we have $K \leq N$. Unlike traditional FedAvg-based methods, which directly dispatch the global model to each client, FedPhoenix first duplicates the global model into $K$ copies and then stochastically resets a few parameters in each copy. Then, the cloud server dispatches the model copies to activated clients, where each client uses its received copy and local data to conduct local training. After local training, the cloud server collects all local models and aggregates them to generate a new global model. Specifically, the workflow of each FedPhoenix training round consists of five steps as follows:

**Step 1: Parameter Resetting.** The cloud server duplicates the global model into $K$ independent copies and stochastically resets a subset of each copy's parameters. The reset proportion of each layer is dynamically adjusted based on the initial value $\mu_0$ and the current training round $r$.

**Step 2: Model Dispatching.** The server sends the reset model copies to $K$ activated clients, where each client only receives one copy.

**Step 3: Local Training.** Activated clients perform model training using their local data.

**Step 4: Model Uploading.** Each activated client uploads its trained local model to the cloud server.

**Step 5: Model Aggregation.** The cloud server aggregates all the uploaded local models to generate a new global model, which is used for the next round of FL training.

### 4.2 Implementation

Algorithm 1 presents the implementation of FedPhoenix. As shown in Algorithm 1, Line 1 initializes the global model $w_{glb}^0$. Lines 2-12 show the FL training process, which consists of $round$ training rounds. Line 3 randomly selects $K$ clients from the client pool $C$, where $S_c$ is the set of selected clients. Lines 4-7 duplicate the global model and reset the model copies, where in Line 6, the function Reset($\cdot$) resets the $k^{th}$ model copy, and such model will be dispatched to the $k^{th}$ activated client in $S_c$. Lines 8-10 present the local training process, where in Line 9, each activated client uses their local data to train the dispatched model $w_k^{Local}$, and $w_k^r$ is the trained local model. Line 11 aggregates all the trained local models to generate the new global model $w_{glb}^r$.

---

**Algorithm 1** FedPhoenix Framework

**Input:** i) $round$, # of training rounds; ii) $C$, the set of involved clients; iii) $K$, # of activated clients; iv) $r_s$, # of rounds for reset; v) $\theta$, reset proportion
**Output:** $w_{glb}$, the trained global model.
**FedPhoenix**($round$,$C$,$K$,$r_s$,$\mu_0$)
1: $w_{glb}^0 \leftarrow$ Initialize global model
2: **for** $r = 1, ..., round$ **do**
3:    $S_c \leftarrow$ Randomly select $K$ clients from $C$
4:    **for** $k = 1, ..., K$ **do**
5:       $w_k^{Local} \leftarrow w_{glb}^{r-1}$
6:       $w_k^{Local} \leftarrow$ Reset($w_k^{Local}, \theta, r, r_s$)
7:    **end for**
8:    */* parallel for block */*
9:    **for** $k = 1, ..., K$ **do**
10:      $w_k^r \leftarrow$ LocalTraining($w_k^{Local}, S_c[k]$)
11:    **end for**
12:    $w_{glb}^r \leftarrow \frac{1}{K} \sum_{k=1}^{K} w_k^r$
13: **end for**
14: **return** $w_{glb}^r$

---

#### 4.2.1 Parameter Reset

Algorithm 2 details the model reset process. As shown in Algorithm 2, Line 2 extracts all the convolutional layers in the target model $w'$. Lines 3-13 perform the parameter reset for the target model $w'$. Line 5 adopts a dynamic stabilization strategy to determine whether to reset the current layer according to the index of the current layer $i$, the current round $r$, and the threshold $r_s$. Line 7 calculates the number of reset kernels according to the reset portion $\mu_0$. Line 8 randomly selects $k_i$ kernels in $l_i$ for reset. Line 10 resets the parameters of the selected kernel, where $w'_{l_i}[k]$ denotes the corresponding parameters of the kernel $k$ in the $w'$. Here, we use a normal distribution to sample from the current layer's mean and variance, and use the sampled values to reset the parameters rather than setting them to zero. The sampling process can be defined as follows:

$$Sampling(l_i) \sim \mathcal{N}(\mu_i, \sigma_i^2), \qquad (1)$$

---

**Algorithm 2** Model Reset

**Input:** i) $w$, model weights; ii) $\theta$, reset proportion; iii) $r$, current round; iv) $r_s$, # of rounds for reset;
**Output:** $w'$, reset model weights.
**Reset**$(w, \mu_0, r, r_s)$
1: $w' \leftarrow w$
2: $S_l \leftarrow$ Convolutional layers in $w'$
3: **for** $i = 1, 2, ..., |S_l|$ **do**
4:      $l_i \leftarrow S_l[i]$
5:      **if** $r \leq \frac{i \times r_s}{|S_l|}$ **then**
6:         $n_i \leftarrow$ number of kernels in layer $l_i$
7:         $k_i \leftarrow \lfloor \theta \times n_i \rfloor$
8:         $S_k \leftarrow$ Randomly select $k_i$ kernels in $l_i$
9:         **for** $k \in S_k$ **do**
10:           $w'_{l_i}[k] \leftarrow Sampling(l_i)$
11:         **end for**
12:      **end if**
13: **end for**
14: **return** $w'$

---

where $\mu_i$ and $\sigma_i$ denote the mean and standard deviation of the parameters of $l_i$, respectively. This design limits the range of resets and satisfies the bounded-variance requirements that ensure the convergence of FedPhoenix.

#### 4.2.2 Dynamic Stabilization Strategy

FedPhoenix employs a dynamic stabilization strategy to maintain stable training and convergence by adjusting the number of reset layers. Initially, more parameters are reset to promote the acquisition of generalized features. As training progresses, the frequency of parameter resets decreases, enabling the model to stabilize its learned features. Since each neural network layer relies on the output of the previous layer, this strategy progressively reduces resets in shallower layers. As shown in the Line 5 of Algorithm 2, assume a network with $|S_l|$ convolutional layers and $l_i$ is the $i^{th}$ convolutional layers, when the current round $r \leq \frac{i \times r_s}{|S_l|}$, the cloud server will perform parameter reset for $l_i$, where $r_s$ is a hyperparameter. However, under the dynamic stabilization strategy, as the number of model layers participating in Reset gradually decreases, FedPhoenix still maintains strong accuracy in scenarios with a large number of clients.

## 5 Convergence Analysis

To demonstrate the convergence of our FedPhoenix approach (assuming that the entire equipment is involved), we present the corresponding convergence analysis. Inspired by the convergence analysis of FedAvg [37], we present the following assumptions:

**Assumption 1.** *For $i \in \{1, 2, \cdots, K\}$, $f_i$ is L-smooth, where*

$$\|\nabla F_k(\mathbf{w}) - \nabla F_k(\mathbf{w}')\| \leq L\|\mathbf{w} - \mathbf{w}'\| \quad \forall \mathbf{w}, \mathbf{w}'.$$

**Assumption 2.** *The variance of stochastic gradients is bounded:*

$$\mathbb{E}_\xi \|\nabla F_k(\mathbf{w}; \xi) - \nabla F_k(\mathbf{w})\|^2 \leq \sigma^2 \quad \forall k, \mathbf{w}.$$

**Assumption 3.** *The expected squared norm of gradients is bounded:*

$$\mathbb{E}\|\nabla F_k(\mathbf{w}; \xi)\|^2 \leq G^2 \quad \forall k, \mathbf{w}.$$

**Assumption 4.** *The divergence between local and global objectives is bounded:*

$$\frac{1}{K} \sum_{k=1}^{K} \|\nabla F_k(\mathbf{w}) - \nabla F(\mathbf{w})\|^2 \leq \beta^2 \quad,$$

Based on the above assumptions and Lemmas 1 and 2 in B.2, we have Theorem 5.1:

**Theorem 5.1** (Convergence Guarantee). *The FedPhoenix algorithm achieves the following asymptotic convergence rate:*

$$\lim_{r \to r_s} D_\theta = 0, \tag{2}$$

$$\mathbb{E}[F(w_T)] - F^* \leq \frac{\kappa}{\gamma + T} \left( \frac{2B}{\mu} + \frac{\mu\gamma}{2} \mathbb{E}\|w_1 - w^*\|^2 \right), \tag{3}$$

matching vanilla FedAvg's rate while enabling transient exploration. Please refer to Appendix B for the full proof.

## 6 Experiments

### 6.1 Experimental Settings

To validate FedPhoenix under diverse data heterogeneity scenarios, we partition each dataset via IID sampling and non-IID Dirichlet Distribution [38]. By default, we assumed that only 10% of the clients participate in each FL communication round. For all experiments, we employed the SGD optimizer with a learning rate of 0.01 and a momentum of 0.5. Each FL training round utilizes a batch size of 50 and 5 local training epochs. For data heterogeneity, we used the Dirichlet distribution [39] $Dir(\beta)$, where $\beta$ is a hyperparameter to control the degree of data heterogeneity. Note that a smaller $\beta$ indicates a higher degree of data heterogeneity. We selected five baseline methods, i.e., FedAvg [1], FedProx [40], FedGen [23], ClusteredSampling [28], and FedMut [26]. Here, FedAvg is the most classical FL method, while the other four methods are SOTA FL methods. Specifically, FedProx is a global control variable-based method, FedGen is a KD-based approach, ClusteredSampling is a device grouping-based method, and FedMut is a mutation-based method. For FedPhoenix, we set the reset rate to $\theta = \frac{1}{32}$ for both IID and non-IID scenarios with $\beta = 0.6$, and set $\theta = \frac{1}{64}$ for non-IID scenarios with $\beta = 0.3$. All the experimental results were obtained from an Ubuntu workstation with an Intel i9 CPU, 256GB of memory, and an NVIDIA RTX 4090 GPU.

Table 2: Test Accuracy Comparison for Both Non-IID and IID Scenarios Using Three DL Models

| Model | Dataset | Hetero Settings | Accuracy of Different Approaches (%) | | | | | |
|---|---|---|---|---|---|---|---|---|
| | | | FedAvg | FedProx | FedMut | CluSample | FedGen | FedPhoenix |
| Resnet | CIFAR-10 | $\beta = 0.3$ | 58.92±0.32 | 58.80±0.39 | 64.04±0.33 | 58.06±0.46 | 60.31±0.23 | **80.12±0.54** |
| | | $\beta = 0.6$ | 64.44±0.15 | 65.11±0.04 | 67.78±0.09 | 63.55±0.19 | 65.56±0.15 | **82.28±0.82** |
| | | *IID* | 64.12±0.03 | 64.75±0.17 | 67.61±0.02 | 64.85±0.01 | 64.47±0.12 | **84.85±0.07** |
| | CIFAR-100 | $\beta = 0.3$ | 41.45±0.13 | 40.23±0.41 | 44.45±0.08 | 39.60±0.16 | 40.82±0.24 | **56.58±0.20** |
| | | $\beta = 0.6$ | 42.98±0.08 | 43.88±0.07 | 46.53±0.02 | 43.88±0.10 | 42.13±0.12 | **58.86±0.17** |
| | | *IID* | 42.97±0.05 | 42.37±0.16 | 44.68±0.29 | 42.59±0.14 | 42.59±0.19 | **62.51±0.01** |
| | Tiny-ImageNet | $\beta = 0.3$ | 48.77±0.10 | 48.52±0.12 | 50.51±0.04 | 48.46±0.11 | 48.10±0.06 | **59.04±0.03** |
| | | $\beta = 0.6$ | 48.98±0.12 | 48.73±0.08 | 51.05±0.05 | 49.45±0.10 | 48.87±0.05 | **59.74±0.09** |
| | | *IID* | 49.56±0.05 | 49.18±0.05 | 50.56±0.03 | 49.46±0.04 | 48.26±0.04 | **60.96±0.12** |
| VGG | CIFAR-10 | $\beta = 0.3$ | 75.92±1.16 | 77.33±0.10 | 79.62±0.47 | 75.36±0.13 | 76.88±0.52 | **81.46±0.70** |
| | | $\beta = 0.6$ | 79.17±0.06 | 78.23±0.09 | 80.45±0.19 | 78.16±0.37 | 79.64±0.05 | **83.24±0.35** |
| | | *IID* | 80.85±0.01 | 80.16±0.01 | 81.46±0.01 | 79.98±0.01 | 80.47±0.00 | **87.17±0.00** |
| | CIFAR-100 | $\beta = 0.3$ | 55.02±0.05 | 54.89±0.37 | 56.81±0.50 | 54.56±0.54 | 54.53±0.74 | **58.84±0.07** |
| | | $\beta = 0.6$ | 56.40±0.10 | 56.97±0.51 | 57.88±0.34 | 56.89 ± 0.58 | 55.42±0.10 | **59.85±0.25** |
| | | *IID* | 58.13±0.01 | 58.59±0.01 | 59.40±0.01 | 58.53±0.08 | 56.54±0.08 | **61.00±0.12** |
| | Tiny-ImageNet | $\beta = 0.3$ | 44.62±0.06 | 44.28±0.11 | 47.05±0.04 | 44.58±0.06 | 51.35±0.13 | **56.89±0.06** |
| | | $\beta = 0.6$ | 45.21±0.03 | 45.13±0.12 | 47.94±0.06 | 50.89±0.02 | 51.45±0.12 | **56.99±0.11** |
| | | *IID* | 45.04±0.01 | 44.85±0.11 | 47.00±0.02 | 51.86±0.21 | 51.89±0.09 | **58.19±0.02** |
| MobNet-V1 | CIFAR-10 | $\beta = 0.3$ | 50.71±3.79 | 51.98±1.47 | 56.22±1.10 | 50.29±5.49 | 51.39±5.32 | **73.54±0.94** |
| | | $\beta = 0.6$ | 57.96±1.06 | 59.89±0.63 | 62.21±0.07 | 60.84±0.47 | 58.80±0.64 | **78.95±0.33** |
| | | *IID* | 60.91±0.13 | 61.18±0.15 | 64.40±0.06 | 62.73±0.15 | 61.03±0.01 | **82.50±0.02** |
| | CIFAR-100 | $\beta = 0.3$ | 35.62±0.26 | 36.18±0.22 | 38.49±0.14 | 36.15±0.31 | 36.05±0.15 | **51.77±1.00** |
| | | $\beta = 0.6$ | 39.32±0.05 | 39.68±0.08 | 40.75±0.08 | 39.23±0.10 | 38.09±0.10 | **53.13±0.14** |
| | | *IID* | 41.50±0.22 | 40.97±0.15 | 46.49±0.18 | 40.89±0.06 | 39.67±0.09 | **57.35±0.11** |
| | Tiny-ImageNet | $\beta = 0.3$ | 39.99±0.06 | 39.97±0.07 | 42.42±0.05 | 39.99±0.05 | 40.12±0.07 | **53.74±0.11** |
| | | $\beta = 0.6$ | 41.15±0.04 | 41.20±0.06 | 43.69±0.04 | 41.46±0.03 | 41.15±0.05 | **54.87±0.04** |
| | | *IID* | 40.78±0.01 | 41.05±0.03 | 43.54±0.02 | 40.27±0.07 | 40.85±0.04 | **57.14±0.07** |

### 6.2 Performance Comparison

Table 2 compares classification performance between FedPhoenix and the baseline models, evaluated under both IID and non-IID conditions, across different datasets and Deep Learning (DL) mod-

els. From Table 2, we observe that FedPhnoeix achieves the best performance across all cases. We observe that FedPhoenix achieves significant accuracy improvements in both IID and two non-IID scenarios, demonstrating its effectiveness in data-heterogeneous settings. We can see that, compared to cases using the VGG-16 model, FedPhoenix achieves greater improvement on the ResNet-18 and MobileNet-V1 models. This is because our parameter reset strategy mainly operates on feature extractors. FedPhoenix performs better on networks that primarily rely on convolutional layers than on dense networks like VGG-16, which have more fully connected layers. It is important to note that FedPhoenix continues to outperform baseline models by a significant margin when using VGG-16.

## 6.3 Compatibility Analysis

**Impact of Different Number of Activated Clients.** We conducted experiments to examine the impact of different numbers of activated clients (i.e., $K \in \{1, 10, 20, 50, 100\}$) on FedPhoenix. Figure 5 compares the training performance between FedPhoenix and the five baselines, considering different numbers of activated clients. We observe that FedPhoenix still significantly outperforms all baselines across all cases. In addition, we observe that when fewer devices are activated, FedPhoenix exhibits significant fluctuations in accuracy during the early training stage, yet it still converges as training progresses.

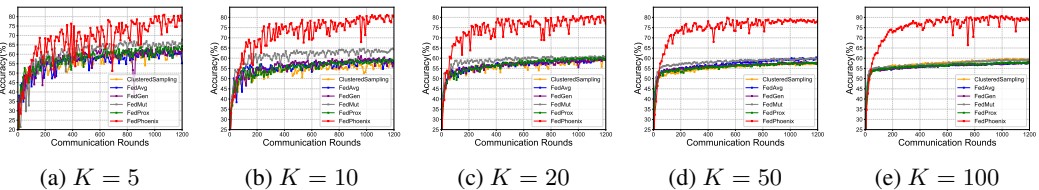

| (a) $K = 5$ | (b) $K = 10$ | (c) $K = 20$ | (d) $K = 50$ | (e) $K = 100$ |

Figure 5: Learning curves on ResNet-18 for different number of active clients with $\alpha = 0.3$

**Impact of the Total Number of Clients.** To explore the impact of the total number of clients, we conducted experiments with different configurations of the total number of clients (i.e., $|C| \in \{50, 100, 200, 500\}$). Figure 6 presents the learning curves of FedPhoenix and five baselines with configurations of the number of clients. We observe that FedPhoenix still significantly outperforms all baselines across all cases. We can still observe that when $|C| = 500$, FedPhoenix should spend more training rounds to achieve the highest accuracy. This is mainly because, as the number of clients increases, the amount of data available per client decreases, inherently limiting the quality of updates each client can contribute. Nonetheless, FedPhoenix can maintain comparable accuracy even with a large number of clients.

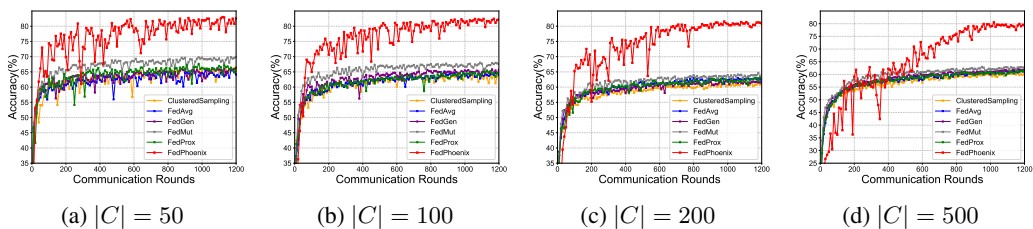

| (a) $|C| = 50$ | (b) $|C| = 100$ | (c) $|C| = 200$ | (d) $|C| = 500$ |

Figure 6: Learning curves for different number of clients on CIFAR-10 dataset with $\beta = 0.6$

**Compatibility with General Overfitting Mitigation Methods.** To evaluate the compatibility of FedPhoenix with general overfitting mitigation methods, we conducted experiments on FedPhoenix and five baselines using two overfitting mitigation techniques in local training, all based on the CIFAR-100 dataset. First, we added Dropout layers after each of the four blocks of ResNet18 and before the fully connected layer, with a dropout probability of $p = 0.5$, as suggested in the original Dropout paper [41]. The results, shown in Table 3, indicate that FedPhoenix achieves the best performance in both IID and Non-IID scenarios. In both Non-IID (with $\beta = 0.3$) and IID settings, FedPhoenix shows a performance discrepancy of merely 2.41%, in contrast to the baselines, which range from 9.43% to 15.96%. This highlights its strong compatibility with Dropout and its superior adaptability to Non-IID conditions. Additionally, we evaluated the performance of ResNet-18 with weight decay, set to 0.0004. This value was chosen based on recommendations from [42]. As shown

Table 3: Test Accuracy Comparison for ResNet-18 with Dropout Layers on CIFAR-100

| Hetero | FedAvg | FedProx | FedMut | CluSamp | FedGen | FedPhoenix |
|--------|--------|---------|--------|---------|--------|------------|
| $\beta$=0.3 | 45.81±1.16 | 48.24±1.02 | 53.42±2.39 | 43.97±1.46 | 49.41±1.18 | **69.10±0.35** |
| $\beta$=0.6 | 54.43±0.29 | 52.76±0.41 | 53.46±0.66 | 54.19±0.74 | 58.15±0.80 | **69.03±0.08** |
| IID | 58.19±0.21 | 57.67±0.39 | 66.87±0.13 | 59.93±0.18 | 62.51±0.09 | **71.51±0.05** |

Table 4: Test Accuracy Comparison for ResNet-18 with a Weight Decay of 0.0004 on CIFAR-100

| Hetero | FedAvg | FedProx | FedMut | CluSamp | FedGen | FedPhoenix |
|--------|--------|---------|--------|---------|--------|------------|
| $\beta$=0.3 | 41.73±0.41 | 41.23±0.35 | 44.59±0.25 | 41.75±0.09 | 43.10±0.09 | **56.73±0.45** |
| $\beta$=0.6 | 44.07±0.20 | 43.92±0.14 | 47.99±0.11 | 43.58±0.06 | 44.26±0.17 | **59.17±0.06** |
| IID | 43.10±0.02 | 42.91±0.03 | 47.18±0.08 | 43.64±0.03 | 43.95±0.03 | **63.08±0.06** |

in Table 4, FedPhoenix consistently outperforms others in both IID and Non-IID settings, further confirming its compatibility with general overfitting mitigation methods.

## 6.4 Ablation Studies

**Impacts of Hyper-parameters.** We performed ablation studies to analyze the effects of hyperparameters on FedPhoenix, specifically examining different reset rounds $r_s$ and reset rates $\theta$ on the CIFAR-10 dataset in a non-IID setting ($\beta = 0.3$) with MobileNet-V1. Figure 7(a) presents the learning curves of FedPhoenix with different values of $r_s$, where "FedPhoenix-1000" denotes the variant of FedPhoenix with $r_s = 1000$. We can observe that a small value of $r_s$ results in performance degradation of FedPhoenix. This is because a few rounds of resetting result in the model not yet learning generalized features. In addition, we can find that as $r_s$ increases, the performance of FedPhoenix improves. Figure 7(b) presents the learning curves of FedPhoenix with different values of $\theta$. We can find that a large value of $\theta$ results in significant fluctuations in accuracy and a small value of $\theta$ results in the accuracy degradation. However, all the variants of FedPhoenix will eventually converge and still outperform all the baselines.

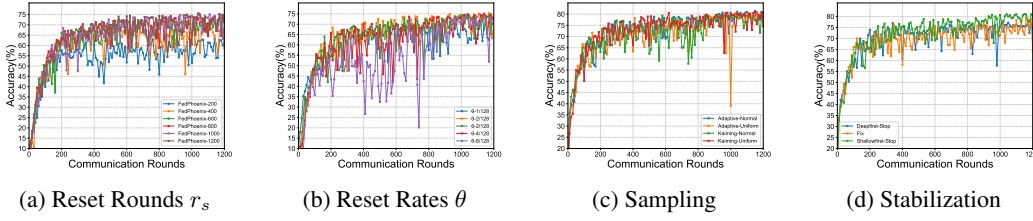

(a) Reset Rounds $r_s$     (b) Reset Rates $\theta$     (c) Sampling     (d) Stabilization

Figure 7: Learning curves of different ablation studies on CIFAR-10 with $\beta = 0.3$

**Impacts of Sampling and Stabilization Strategies.** To evaluate the effectiveness of FedPhoenix's sampling and stabilization strategies, we conducted ablation studies on the CIFAR-10 dataset under a non-IID condition (with $\beta = 0.3$) using ResNet-18 as our model. We compared our sampling strategy to the Kaiming normal distribution (a.k.a., Kaiming-Normal). Figure 7(c) demonstrates that our sampling strategy outperforms the Kaiming-Normal approach. Additionally, Figure 7(d) displays the learning curves for FedPhoenix alongside a variant that does not employ the stabilization strategy. The variant without stabilization exhibited greater fluctuations and lower accuracy than the full FedPhoenix implementation.

## 6.5 Computational Overhead Analysis

To provide a comprehensive view of our method's efficiency, we analyzed its theoretical computational overhead in terms of Floating Point Operations (FLOPs). This analysis shows that our server-side reset mechanism incurs a modest and manageable cost.

**Key Assumptions and Our Method's FLOPs.** Our analysis is based on the following assumptions: computing the layer-wise mean $\mu_i$ and variance $\sigma_i^2$ requires approximately 2 and 4 FLOPs per parameter, respectively. Sampling new parameters is conservatively estimated to take around 10 FLOPs per parameter. The additional overhead of our method occurs entirely on the server side. This includes a one-time computation of statistics, which is estimated at about $6M$ FLOPs (where $M$ represents the total number of model parameters), and a per-client reset operation, estimated at

Table 5: Comparison of per-round computational overhead ($\Delta$FLOPs) relative to vanilla FedAvg. Our method involves a modest server-only cost, unlike methods that require significant client-side computation or struggle to scale with the total number of clients $N$.

| Method | Overhead Source & Location | Per-Round $\Delta$FLOPs |
|---|---|---|
| **FedPhoenix (Ours)** | Server: Statistics computation & parameter sampling for $K$ clients. | $M(6 + 10\theta K)$ |
| FedProx | Client: Proximal term computation in each of the $P$ local steps. | $\approx 3PKM$ |
| FedMut | Server: Global gradient aggregation & model mutation for $K$ clients. | $M(1 + 2K)$ |
| ClusteredSampling | Server: Pairwise similarity computation between $K$ models and all $N$ clients. | $\approx 3KNM$ |
| FedGen | Both: Multi-step generator training (Server) & knowledge distillation (Client). | High* |

*Notations:* $M$, model parameters; $K$, the number of participating clients per round; $N$, the number of total clients ($N \geq K$); $P$, the number local update steps; $\theta$, perturbation ratio.
*FedGen's overhead involves multiple forward/backward passes on both the client and the server, often exceeding those of other methods by orders of magnitude.

approximately $10\theta M$ FLOPs for each client, with $\theta$ being the perturbation ratio and $K$ referring to the number of clients. Consequently, the total overhead per round is given by $\mathbf{M(6 + 10\theta K)}$.

**Comparison with Baselines.** Table 5 compares the additional computational overhead per round ($\Delta$FLOPs) of our method with the baseline methods. This comparison demonstrates that our server-only overhead is highly competitive, as it avoids the significant client-side costs and the poor scalability associated with the total number of clients ($N$) observed in other methods. The results show a clear efficiency advantage for our approach. For a ResNet-18 model ($M \approx 11M$) with $K = 10$ and $\theta = 0.2$, our total reset FLOPs are $\approx 286M$. On an NVIDIA A100 GPU, this adds a negligible wall-clock time of less than 0.1 seconds per round. This is significantly more efficient than methods like FedProx or FedGen, which impose heavy computational burdens on resource-constrained clients, and scales better than ClusteredSampling in scenarios with a large pool of total clients $N$.

## 6.6 Limitations

While the above experiment results are promising, certain limitations point to areas for future research. First, our methods and experiments primarily focus on vision tasks and Convolutional Neural Networks (CNNs). It remains uncertain whether the FedPhoenix framework can be applied to other domains, such as Natural Language Processing (NLP) tasks and Transformer architectures. We plan to adapt the FedPhoenix framework for these models and tasks in our future studies. Additionally, the convergence analysis presented here is limited to scenarios where all devices participate fully. The stage-wise perturbation decay mechanism indicates that the perturbation resulting from the reset operation will eventually decay to zero, even with partial participation. This could lead to convergence behavior similar to that of FedAvg; however, this theory needs comprehensive theoretical analysis and experimental validation.

## 7 Conclusion

This paper introduces *FedPhoenix*, a novel FL framework that incorporates a heuristic parameter reset strategy to improve the generalization of the global model in non-IID scenarios. By randomly resetting certain subsets of global model parameters, FedPhoenix disrupts client-specific features and promotes collaborative restoration of more generalized features across clients. Theoretical analysis shows that FedPhoenix converges effectively, and experiments conducted on three benchmark datasets demonstrate its effectiveness.

## Acknowledgment

This work was supported by the Natural Science Foundation of China (62272170), the Shanghai International Joint Lab of Trustworthy Intelligent Software (22510750100), and the National Research Foundation, Singapore, and Cyber Security Agency of Singapore under its National Cybersecurity R&D Programme and CyberSG R&D Cyber Research Programme Office. Any opinions, findings and conclusions or recommendations expressed in this material are those of the author(s) and do not reflect the views of National Research Foundation, Singapore, Cyber Security Agency of Singapore as well as CyberSG R&D Programme Office, Singapore.

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

# A Additional Experimental Results

## A.1 Learning Curves of Different FL Methods

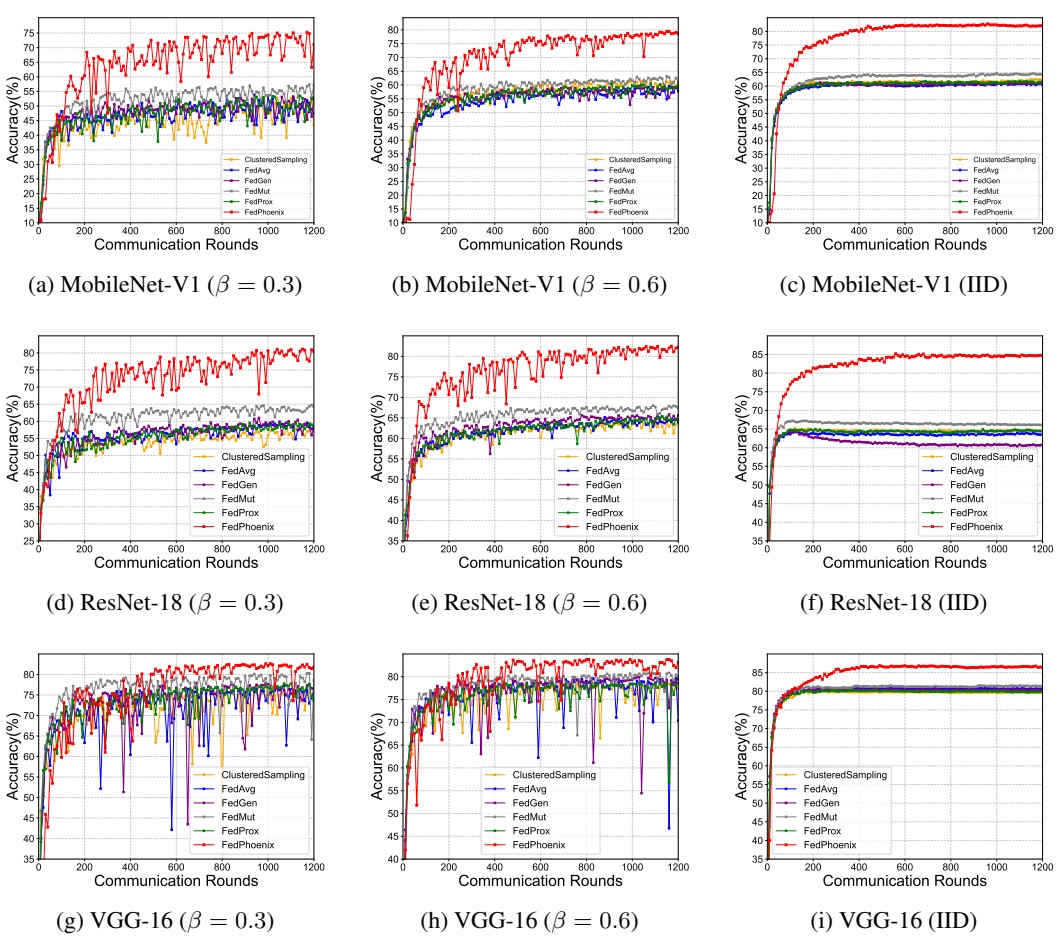

(a) MobileNet-V1 ($\beta = 0.3$)    (b) MobileNet-V1 ($\beta = 0.6$)    (c) MobileNet-V1 (IID)

(d) ResNet-18 ($\beta = 0.3$)    (e) ResNet-18 ($\beta = 0.6$)    (f) ResNet-18 (IID)

(g) VGG-16 ($\beta = 0.3$)    (h) VGG-16 ($\beta = 0.6$)    (i) VGG-16 (IID)

Figure 8: Learning curves comparison on CIFAR-10 with three network architectures under different data distributions. From top to bottom: MobileNet-V1, ResNet-18, and VGG-16. Columns show different Dirichlet concentration parameters ($\beta = 0.3, 0.6$) and an IID setting

## A.2 Supplementary Tables of Test Accuracy Comparison (Extended from Section 6.3)

We evaluated the performance of FedPhoenix and baseline methods using a ResNet-18 architecture with consistent regularization measures (e.g., Dropout and Weight Decay) on both CIFAR-10 and Tiny-ImageNet datasets. Experimental results demonstrate that FedPhoenix consistently outperforms all baselines, achieving state-of-the-art accuracy under identical experimental conditions.

Table 6: Comparison for ResNet-18 with Dropout layers on CIFAR-10 and Tiny-ImageNet

| Dataset | Hetero | FedAvg | FedProx | FedMut | CluSamp | FedGen | FedPhoenix |
|---------|--------|--------|---------|--------|---------|--------|------------|
| CIFAR-10 | $\beta$=0.3 | 59.89±14.77 | 61.07±3.76 | 66.60±0.27 | 58.95±10.05 | 79.51±4.04 | **85.54±0.11** |
| | $\beta$=0.6 | 75.42±3.51 | 73.05±1.61 | 80.23±1.38 | 70.71±9.84 | 85.06±2.35 | **87.56±0.27** |
| | IID | 76.72±0.47 | 78.92±0.21 | 82.37±0.20 | 75.74±0.32 | 89.69±0.05 | **90.46±0.02** |
| Tiny-ImageNet | $\beta$=0.3 | 49.81±0.31 | 50.55±0.10 | 52.04±0.07 | 50.71±0.07 | 50.35±0.09 | **57.68±0.05** |
| | $\beta$=0.6 | 51.03±0.06 | 51.88±0.11 | 52.67±0.07 | 52.05±0.04 | 52.25±0.04 | **59.54±0.04** |
| | IID | 51.78±0.03 | 52.77±0.03 | 53.62±0.02 | 52.26±0.03 | 52.12±0.13 | **61.01±0.02** |

Table 7: Comparison for ResNet-18 with Weight Decay = 0.0004 on CIFAR-10 and Tiny-ImageNet

| Dataset | Hetero | FedAvg | FedProx | FedMut | CluSamp | FedGen | FedPhoenix |
|---------|--------|--------|---------|--------|---------|--------|------------|
| CIFAR-10 | $\beta$=0.3 | 60.25±6.12 | 60.51±0.29 | 64.13±0.40 | 58.58±2.65 | 60.08±2.11 | **79.20±2.51** |
| | $\beta$=0.6 | 66.31±0.18 | 64.42±0.26 | 70.87±0.05 | 63.91±0.22 | 66.54±0.21 | **81.70±0.11** |
| | IID | 65.28±0.03 | 64.75±0.04 | 67.82±0.04 | 65.98±0.02 | 67.38±0.03 | **85.24±0.01** |
| Tiny-ImageNet | $\beta$=0.3 | 51.25±0.10 | 51.01±0.08 | 52.57±0.14 | 50.62±0.13 | 51.30±0.09 | **58.34±0.10** |
| | $\beta$=0.6 | 51.82±0.24 | 51.91±0.14 | 52.92±0.18 | 52.05±0.15 | 52.89±0.08 | **59.30±0.10** |
| | IID | 51.14±0.05 | 51.54±0.04 | 51.93±0.06 | 52.26±0.04 | 52.56±0.01 | **61.29±0.02** |

## A.3 Comparison of Generalization

According to the observations in [43, 44], the more generalized model is typically located in flat areas rather than sharp areas. To evaluate the generalization of FedPhoenix, as shown in Figure 9, we visualized the loss landscapes of the models trained by FedAvg and FedPhoenix on CIFAR-10 under IID and non-IID ($\beta = 0.3$) settings. We can observe that the global model trained by FedAvg is located in a sharp area. In contrast, that trained by FedPhoenix converges to a significantly flatter area across both settings, which demonstrates the good generalization ability of FedPhoenix.

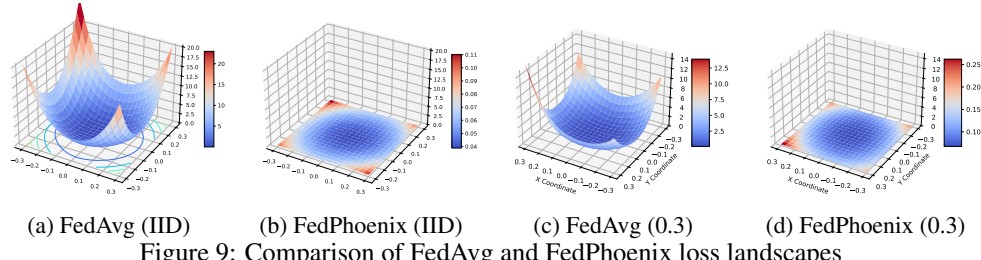

(a) FedAvg (IID)      (b) FedPhoenix (IID)      (c) FedAvg (0.3)      (d) FedPhoenix (0.3)

Figure 9: Comparison of FedAvg and FedPhoenix loss landscapes

## A.4 Performance under Extreme Data Heterogeneity

To thoroughly test the limits of FedPhoenix, we conducted experiments under two challenging scenarios: extreme skew in label distribution and extreme skew in feature distribution.

### A.4.1 Performance under Extreme Label Skew

We evaluated our method against baselines on CIFAR-10 with highly skewed label distributions, simulated by a Dirichlet distribution with a very small concentration parameter ($\beta = 0.01$ and $\beta = 0.1$). As shown in Table 8, FedPhoenix demonstrates significant performance gains across various model architectures (ResNet-18, VGG16, and MobileNet-V1), especially in the most extreme non-IID setting ($\beta = 0.01$). This highlights its robustness and superior ability to mitigate client drift when local data distributions are severely biased.

Table 8: Accuracy (%) on CIFAR-10 under extreme non-IID (label skew) settings. FedPhoenix consistently outperforms all baselines, with particularly strong gains when $\beta = 0.01$.

| Model | Setting ($\beta$) | FedAvg | FedProx | FedMut | CluSamp | FedGen | FedPhoenix (Ours) |
|-------|-------------------|--------|---------|--------|---------|--------|-------------------|
| ResNet-18 | 0.01 | $29.33 \pm 0.30$ | $27.88 \pm 0.41$ | $28.20 \pm 0.33$ | $28.53 \pm 0.26$ | $29.46 \pm 0.04$ | $\mathbf{35.80 \pm 0.93}$ |
| | 0.1 | $45.48 \pm 0.03$ | $45.41 \pm 0.16$ | $47.55 \pm 0.14$ | $45.56 \pm 0.36$ | $45.29 \pm 7.33$ | $\mathbf{55.80 \pm 0.11}$ |
| VGG16 | 0.01 | $32.24 \pm 0.16$ | $31.79 \pm 0.37$ | $32.09 \pm 0.16$ | $30.74 \pm 0.57$ | $32.03 \pm 1.89$ | $\mathbf{33.10 \pm 0.41}$ |
| | 0.1 | $47.75 \pm 0.11$ | $48.26 \pm 0.18$ | $49.59 \pm 0.07$ | $47.72 \pm 0.05$ | $49.30 \pm 0.29$ | $\mathbf{52.86 \pm 0.14}$ |
| MobileNet-V1 | 0.01 | $22.24 \pm 0.16$ | $21.79 \pm 0.37$ | $22.09 \pm 0.16$ | $20.74 \pm 0.57$ | $21.01 \pm 1.89$ | $\mathbf{27.56 \pm 0.57}$ |
| | 0.1 | $47.75 \pm 0.11$ | $48.26 \pm 0.18$ | $49.59 \pm 0.07$ | $47.72 \pm 0.05$ | $37.93 \pm 0.29$ | $\mathbf{50.36 \pm 0.11}$ |

### A.4.2 Performance on Extreme Feature Skew (Office-31 Dataset)

To simulate a scenario where clients' feature spaces are disjoint, we used the Office-31 dataset, a benchmark for domain adaptation. We assigned each of the three domains (i.e., Amazon, DSLR, Webcam) to a separate client. This setup mimics a federated environment with extreme feature skew. The results in Table 9 show that FedPhoenix achieves a substantial accuracy improvement over all baselines. This empirically validates that our method effectively facilitates collaboration and knowledge transfer even when clients' data come from entirely different distributions.

Table 9: Accuracy (%) on the Office-31 dataset, partitioned by domain to simulate extreme feature skew. FedPhoenix significantly outperforms other methods.

| Dataset | FedAvg | FedProx | FedMut | CluSamp | FedGen | FedPhoenix (Ours) |
|---------|--------|---------|--------|---------|--------|-------------------|
| Office-31 | $47.84 \pm 1.78$ | $48.12 \pm 1.68$ | $43.99 \pm 1.59$ | $46.42 \pm 1.83$ | $49.13 \pm 2.26$ | $\mathbf{54.17 \pm 0.96}$ |

## A.5   Ablation Studies on the Reset Operation

We conducted a series of ablation studies to justify the core design choices of our reset operations.

### A.5.1   Target Layers: Convolutional vs. Fully-Connected

To validate our rationale for exclusively targeting Convolutional (Conv) layers, we compared it with applying the Reset Operation to Fully-Connected (FC) layers only or to both. The results in Table 10 confirm our hypothesis. Resetting only Conv layers yields the best or highly competitive performance, especially in the non-IID setting. Conversely, resetting FC layers alone is detrimental, causing a significant drop in accuracy. This supports our design choice to enhance the shared feature extractor (Conv layers) while preserving the stability of the classifier (FC layers).

Table 10: Ablation study on the target layers for the Reset Operation (ResNet and VGG on CIFAR-10). Results show that resetting only convolutional layers is the most effective strategy.

| Model | Setting | Both Conv & FC | Conv Layers Only (Ours) | FC Layers Only |
|-------|---------|----------------|-------------------------|----------------|
| ResNet | Non-IID ($\beta = 0.3$) | $78.56 \pm 1.08$ | $\mathbf{80.12 \pm 0.54}$ | $57.17 \pm 0.71$ |
|  | IID | $\mathbf{85.31 \pm 0.01}$ | $85.06 \pm 0.03$ | $66.55 \pm 0.03$ |
| VGG | Non-IID ($\beta = 0.3$) | $\mathbf{82.21 \pm 0.28}$ | $81.46 \pm 0.70$ | $76.81 \pm 0.27$ |
|  | IID | $\mathbf{87.17 \pm 0.00}$ | $\mathbf{87.17 \pm 0.00}$ | $78.92 \pm 0.03$ |

### A.5.2   Reset Granularity: Full vs. Partial Kernel Reset

We investigated whether resetting an entire kernel is superior to resetting only a fraction of its most significant parameters (by magnitude). We compared our full-kernel reset with partially resetting the top 1/3 or top 2/3 parameters within each selected kernel. As shown in Table 11, the original FedPhoenix approach of resetting the entire kernel as a single functional unit consistently outperforms partial resets. We hypothesize that a full reset more effectively promotes feature exploration and diversity.

Table 11: Ablation study on reset granularity (Non-IID CIFAR-10, ResNet-18). FedPhoenix's full-kernel reset is compared against resetting only the top-magnitude parameters within a kernel.

| Setting ($\beta$) | Top-1/3 Reset | Top-2/3 Reset | Top-1/3 (Fixed $\theta$) | Top-2/3 (Fixed $\theta$) | FedPhoenix (Full Kernel) |
|-------------------|---------------|---------------|--------------------------|--------------------------|--------------------------|
| 0.3 | $77.20 \pm 1.87$ | $78.47 \pm 1.32$ | $76.86 \pm 0.98$ | $79.55 \pm 0.95$ | $\mathbf{80.12 \pm 0.54}$ |
| 0.6 | $80.68 \pm 0.17$ | $80.90 \pm 1.06$ | $80.35 \pm 0.05$ | $81.76 \pm 0.78$ | $\mathbf{82.28 \pm 0.82}$ |
| IID | $83.01 \pm 0.01$ | $84.78 \pm 0.02$ | $84.26 \pm 0.01$ | $84.63 \pm 0.01$ | $\mathbf{84.85 \pm 0.07}$ |

### A.5.3   Sampling Distributions for Kernel Initialization

To test the robustness of our method to the choice of sampling distribution, we conducted an extensive study comparing our proposed adaptive Normal distribution (`Adp_Normal`) with a wide range of other common initialization methods. All distributions were adapted using the target layer's statistics. The results in Table 12 show that FedPhoenix performs consistently well across all tested distributions. This demonstrates that the core strength of our method lies in the adaptive reset strategy itselfre-sampling based on the layer's current statistical propertiesrather than a dependency on a specific distribution.

Table 12: Ablation study on sampling distributions for the Reset Operation (ResNet-18, CIFAR-10). Performance is robust across a wide variety of distributions.

| Non-IID Level | Adp_Normal | Adp_Uniform | Kaiming_Normal | Kaiming_Uniform | Truncated_Normal | Xavier_Normal | Xavier_Uniform | Orthogonal | Laplace |
|---|---|---|---|---|---|---|---|---|---|
| $\beta = 0.3$ | **80.12 ± 0.54** | 80.02 ± 0.41 | 79.34 ± 0.64 | 77.58 ± 2.34 | 79.42 ± 1.45 | 79.47 ± 0.59 | 78.87 ± 2.01 | 80.00 ± 0.16 | 78.46 ± 0.34 |
| $\beta = 0.6$ | 82.28 ± 0.82 | 81.86 ± 0.24 | 79.97 ± 0.17 | 80.34 ± 0.41 | 81.70 ± 0.09 | 80.60 ± 0.21 | 81.03 ± 0.60 | **82.68 ± 0.11** | 81.74 ± 0.07 |
| IID | 85.06 ± 0.03 | 85.25 ± 0.04 | 84.70 ± 0.01 | 84.52 ± 0.03 | 85.08 ± 0.02 | 85.03 ± 0.02 | 84.75 ± 0.01 | **85.42 ± 0.03** | 85.00 ± 0.02 |

# B  Convergence Analysis

## B.1  Notations and Perturbation Mechanism

Let $t$ index the local SGD iteration, and let $v_t$ denote the intermediate model after one SGD step. Let $T = n \times E$ represent the total number of iterations. For layer $l_i$ with $n_i$ kernels, define:

- $|S_l|$ represents the total number of feature layers.

- Perturbation ratio $\theta \in (0,1)$, resetting $\lfloor \theta n_i \rfloor$ kernels per round.

- $\mu_i = \mathbb{E}[W_{l_i}]$, $\sigma_i^2 = \text{Var}(W_{l_i})$ for layer parameters, where $\mu_i$ and $\sigma_i^2$ are the statistical mean and variance computed from the current parameters of layer $l_i$.

- Perturbation operator $\Delta_{l_i} \sim \mathcal{N}(\mu_i, \sigma_i^2)$ for resampled kernels, indicating that new kernel parameters are sampled from a normal distribution with mean $\mu_i$ and variance $\sigma_i^2$.

The layer-wise perturbation variance is derived as follows. For each kernel $j$ (where $j = 1, 2, \ldots, \lfloor \theta n_i \rfloor$) that is reset in layer $l_i$, the new parameter value $w_j^{\text{new}}$ is sampled from $\mathcal{N}(\mu_i, \sigma_i^2)$, while the original parameter value is $w_j$. The perturbation for a single kernel is $w_j^{\text{new}} - w_j$, and the expected squared perturbation is:

$$
\begin{aligned}
\mathbb{E}[(w_j^{\text{new}} - w_j)^2] &= \mathbb{E}[(w_j^{\text{new}} - \mu_i + \mu_i - w_j)^2] \\
&= \mathbb{E}[(w_j^{\text{new}} - \mu_i)^2] + \mathbb{E}[(w_j - \mu_i)^2] + 2\mathbb{E}[(w_j^{\text{new}} - \mu_i)(\mu_i - w_j)] \\
&= \mathbb{E}[(w_j^{\text{new}} - \mu_i)^2] + \mathbb{E}[(w_j - \mu_i)^2] \qquad (4) \\
&= \sigma_i^2 + \mathbb{E}[(w_j - \mu_i)^2]. \qquad (5)
\end{aligned}
$$

Here, $\mathbb{E}[(w_j - \mu_i)^2]$ represents the variance of the original parameter $w_j$ around the layer mean $\mu_i$. Since $\sigma_i^2$ is the statistical variance computed from all parameters in layer $l_i$, we approximate $\mathbb{E}[(w_j - \mu_i)^2] \approx \sigma_i^2$ for each parameter $w_j$ in the layer. This approximation is reasonable due to the large number of parameters in typical neural network layers, where the individual parameter variance is close to the layer-wise statistical variance. Thus, we have:

$$
\mathbb{E}[(w_j^{\text{new}} - w_j)^2] \approx \sigma_i^2 + \sigma_i^2 = 2\sigma_i^2. \qquad (6)
$$

For the entire layer $l_i$, with $\lfloor \theta n_i \rfloor$ kernels being reset, the expected perturbation variance is:

$$
\begin{aligned}
\mathbb{E}\|\Delta_{l_i}\|^2 &= \mathbb{E}\left[ \sum_{j=1}^{\lfloor \theta n_i \rfloor} (w_j^{\text{new}} - w_j)^2 \right] \\
&\approx \sum_{j=1}^{\lfloor \theta n_i \rfloor} 2\sigma_i^2 = 2\lfloor \theta n_i \rfloor \sigma_i^2 \leq 2\theta n_i \sigma_i^2. \qquad (7)
\end{aligned}
$$

## B.2  Key Lemmas

**Lemma 1** (Client Drift Bound). *Following the proof structure of Lemma 3 in [45], we bound the expected client drift as:*

$$
\mathbb{E}\sum_{k=1}^{K} \|w_t^k - w_t\|^2 \leq 4\eta_t^2(E-1)^2 G^2 + 4\theta \sum_{i=1}^{|s_l|} n_i \sigma_i^2. \qquad (8)
$$

*Proof.* Let $\Delta^{(k)} = \sum_{i=1}^{|s_l|} \Delta_{l_i}^{(k)}$ represent the total perturbation applied to client $k$'s model across all layers due to kernel resetting at the start of a communication round. The difference between the local model $w_t^k$ of client $k$ and the global model $w_t$ can be decomposed into contributions from the initial perturbation and the local SGD updates. Specifically, starting from the perturbed initialization, we have:

$$\|w_t^k - w_t\|^2 \le 2\|\Delta^{(k)}\|^2 + 2\left\|\sum_{e=0}^{E-1} \eta_{t+e} \nabla F_k(w_{t+e}^k, \xi_{t+e})\right\|^2. \tag{9}$$

Taking expectations on both sides:

- For the perturbation term, using the layer-wise perturbation variance derived earlier (Equation 7), we have:

$$\mathbb{E}\|\Delta^{(k)}\|^2 = \mathbb{E}\left\|\sum_{i=1}^{|s_l|} \Delta_{l_i}^{(k)}\right\|^2 \le \sum_{i=1}^{|s_l|} \mathbb{E}\|\Delta_{l_i}^{(k)}\|^2 \le \sum_{i=1}^{|s_l|} 2\theta n_i \sigma_i^2 = 2\theta \sum_{i=1}^{|s_l|} n_i \sigma_i^2.$$

- For the local SGD update term, under Assumption 3 (bounded gradient norm), we follow the standard analysis in [45]:

$$\mathbb{E}\left\|\sum_{e=0}^{E-1} \eta_{t+e} \nabla F_k(w_{t+e}^k, \xi_{t+e})\right\|^2 \le 2\eta_t^2 (E-1)^2 G^2.$$

Combining these bounds into Equation 9, we obtain:

$$\mathbb{E}\|w_t^k - w_t\|^2 \le 2 \cdot 2\theta \sum_{i=1}^{|s_l|} n_i \sigma_i^2 + 2 \cdot 2\eta_t^2 (E-1)^2 G^2$$

$$= 4\theta \sum_{i=1}^{|s_l|} n_i \sigma_i^2 + 4\eta_t^2 (E-1)^2 G^2. \tag{10}$$

Summing over all clients $k = 1, 2, \ldots, K$, and noting that the bound holds uniformly for each client, we arrive at the desired result:

$$\mathbb{E}\sum_{k=1}^{K} \|w_t^k - w_t\|^2 \le 4\eta_t^2 (E-1)^2 G^2 + 4\theta \sum_{i=1}^{|s_l|} n_i \sigma_i^2. \tag{11}$$

$\square$

**Lemma 2** (One-Step Descent). *Let $\kappa = L/\mu$. For a learning rate $\eta_t = \frac{2}{\mu(\gamma+t)}$ satisfying the conditions in [45], we have:*

$$\mathbb{E}\|v_{t+1} - w^*\|^2 \le (1 - \eta_t \mu)\mathbb{E}\|w_t - w^*\|^2 + \eta_t^2 \left(B + 4\theta \sum_{i=1}^{|s_l|} n_i \sigma_i^2\right), \tag{12}$$

*where $B = \sum_{k=1}^{N} p_k^2 \sigma_k^2 + 6L\Gamma + 8(E-1)^2 G^2$.*

*Proof.* The proof follows the structure of Lemma 1 in [45], which analyzes the one-step descent of SGD in the federated setting under Assumptions 1 (L-smoothness) and 4 (strong convexity). The key difference in FedPhoenix is the additional perturbation introduced by kernel resetting, which affects the client drift term. From Lemma 1 in the standard FedAvg analysis, the one-step descent bound includes terms related to variance, data heterogeneity, and client drift. Incorporating the modified client drift bound from our Lemma 1, the additional perturbation variance term $4\theta \sum_{i=1}^{|s_l|} n_i \sigma_i^2$ is added to the original bound $B$. Thus, we obtain:

$$\mathbb{E}\|v_{t+1} - w^*\|^2 \le (1 - \eta_t \mu)\mathbb{E}\|w_t - w^*\|^2 + \eta_t^2 \left(B + 4\theta \sum_{i=1}^{|s_l|} n_i \sigma_i^2\right), \tag{13}$$

where $B = \sum_{k=1}^{N} p_k^2 \sigma_k^2 + 6L\Gamma + 8(E-1)^2 G^2$ captures the standard variance and divergence terms from FedAvg. $\square$

**Theorem B.1** (Global Convergence). *With $\gamma > \max\{8\kappa, E\}$ and a decaying perturbation ratio $\theta(r)$, the expected optimality gap of the global model after $T$ iterations satisfies:*

$$\mathbb{E}[F(w_T)] - F^* \leq \frac{\kappa}{\gamma + T} \left( \frac{2(B + 4D_\theta)}{\mu} + \frac{\mu\gamma}{2}\mathbb{E}\|w_1 - w^*\|^2 \right), \tag{14}$$

*where $D_\theta = \theta \sum_{i=1}^{|s_l|} n_i\sigma_i^2$ represents the perturbation variance term.*

*Proof.* The proof follows the standard convergence analysis of FedAvg as in [45], leveraging the one-step descent bound from Lemma 2. Starting from the recursive inequality:

$$\mathbb{E}\|v_{t+1} - w^*\|^2 \leq (1 - \eta_t\mu)\mathbb{E}\|w_t - w^*\|^2 + \eta_t^2 \left( B + 4\theta \sum_{i=1}^{|s_l|} n_i\sigma_i^2 \right), \tag{15}$$

we define $\Delta_t = \mathbb{E}\|w_t - w^*\|^2$ and $\eta_t = \frac{2}{\mu(\gamma+t)}$. By induction and the choice of $\gamma > \max\{8\kappa, E\}$, which ensures $\eta_t \leq \frac{1}{4L}$ and $\eta_t \leq 2\eta_{t+E}$, we can telescope the inequality over $T$ iterations. Noting that $4\theta \sum_{i=1}^{|s_l|} n_i\sigma_i^2 = 4D_\theta$, the additional perturbation term is incorporated into the final bound. Using the $L$-smoothness of $F(\cdot)$, the optimality gap is bounded as:

$$\mathbb{E}[F(w_T)] - F^* \leq \frac{L}{2}\Delta_T \leq \frac{\kappa}{\gamma + T} \left( \frac{2(B + 4D_\theta)}{\mu} + \frac{\mu\gamma}{2}\mathbb{E}\|w_1 - w^*\|^2 \right), \tag{16}$$

where $\kappa = L/\mu$. $\qquad\square$

### B.3 Stage-wise Perturbation Decay

To mitigate the impact of perturbation on convergence in later stages, we adopt a stage-wise decay strategy for the perturbation ratio over communication rounds. For a predefined number of communication rounds $r_s$ during which Reset perturbations are applied, the perturbation ratio $\theta(r)$ at round $r$ is defined as:

$$\theta(r) = \theta_0 \left( 1 - \frac{\left\lfloor \frac{r}{r_s} \cdot |S_l| \right\rfloor}{|S_l|} \right), \tag{17}$$

where $\theta_0$ is the initial perturbation ratio, and $|S_l|$ represents the total number of convolutional layers in the model. This ensures that $\theta(r) \to 0$ within $r_s$ rounds of applying perturbations. Substituting the decaying $\theta(r)$ into Theorem B.1, the perturbation variance term $D_\theta = \theta(r) \sum_{i=1}^{|s_l|} n_i\sigma_i^2$ diminishes over time:

$$\lim_{r \to r_s} D_\theta = 0, \tag{18}$$

$$\mathbb{E}[F(w_T)] - F^* \leq \frac{\kappa}{\gamma + T} \left( \frac{2B}{\mu} + \frac{\mu\gamma}{2}\mathbb{E}\|w_1 - w^*\|^2 \right). \tag{19}$$

This recovers the original FedAvg convergence rate with $\mathcal{O}(1/T)$ dependency as the perturbation effect vanishes.

## C Discussions

### C.1 Privacy Preserving

While the primary objective of FedPhoenix is to enhance model generalization and robustness, its unique parameter reset mechanism also offers notable privacy considerations, even though it does not provide formal guarantees like $(\epsilon, \delta)$-Differential Privacy. The core mechanism, which resets a fraction $(\theta)$ of convolution kernels based on the layer's statistics, introduces stochasticity into the global model. This inherent randomness increases the effort an adversary must expend to successfully carry out attacks such as model inversion, as the model parameters are less deterministic across communication rounds.

Crucially, FedPhoenix is fully compatible with standard privacy-enhancing technologies like secure aggregation. Since the server-side reset operation is based solely on the aggregated global model, secure aggregation protocols can be seamlessly applied to protect individual client updates from being inspected by the server.

Furthermore, we can significantly strengthen privacy protection, particularly against a malicious server, by shifting the reset operation to the client side. In this configuration, each activated client receives the same global model and then independently performs the parameter reset locally. This creates a unique training starting point for each client. Consequently, even a malicious server cannot know the exact initial model from which a client's update was generated, which substantially complicates server-side reverse engineering or inference attacks targeting a specific client's data. As our computational analysis demonstrates, the overhead of the reset operation is minimal, making it entirely feasible for individual clients to bear this cost.

For future work, we plan to quantitatively evaluate these inherent privacy benefits. To achieve formal privacy guarantees, FedPhoenix can be further integrated with techniques such as Differential Privacy, for instance, by adding calibrated noise during the parameter sampling process. This would create a comprehensive framework that balances performance, robustness, and provable privacy for secure deployment in real-world federated learning scenarios.

## C.2 Limitations

Although the FedPhoenix method effectively mitigates overfitting issues introduced by distributed data, this work still has several limitations that warrant further exploration in future research. First, the proposed method and experimental design primarily focus on vision tasks and convolutional neural networks (CNNs), which are commonly studied in federated learning. Consequently, its applicability to Transformer architectures and natural language processing (NLP) tasks remains unverified. In subsequent work, we plan to extend the FedPhoenix framework to accommodate these emerging models and tasks, thereby improving its generalization capability.

Additionally, in the Convergence Analysis section, we only discuss the scenario of full device participation. However, considering the Stage-wise Perturbation Decay mechanism, even in partial device participation scenarios, the perturbation introduced by the reset operation will gradually decay to zero. This suggests that the convergence behavior of FedPhoenix may be similar to that of FedAvg, though this hypothesis requires more detailed theoretical analysis and experimental validation to confirm.

