# OpenReview forum: "Rising from Ashes: Generalized Federated Learning via Dynamic Parameter Reset"
_NeurIPS.cc/2025/Conference — NeurIPS 2025 poster_

### Official Review · Reviewer_NLhY · 2025-06-24

**Clarity:** 4
**Significance:** 3
**Originality:** 3
**Rating:** 5
**Confidence:** 4

**Summary:**

This paper proposes the FedPhoenix framework to address the poor generalization ability of global models. The core idea is to randomly reset a subset of parameters to disrupt certain characteristics of the global model.

**Questions:**

See the weaknesses

**Ethical Concerns:**

["NO or VERY MINOR ethics concerns only"]

**Final Justification:**

I just carefully reviewed your responses to all the reviewers’ comments. My concerns have been largely addressed, and it seems the other reviewers’ issues have also been resolved. Please make sure the final version fills the full 9 pages. I will raise my score to 5. Thanks for your rebuttal.

**Limitations:**

yes

**Quality:**

3

**Strengths And Weaknesses:**

Strengths:
1. The method is concise, simple, and effective.

2. The authors provide code (in the supplementary materials).

3. There is substantial theoretical analysis.

4. The experiments are thorough, conducted on three different models and three datasets — notably including ImageNet. The ablation study is also well-designed.

Weaknesses:

Major:
1. The related work section is not well written. The author categorizes the FL methods in the introduction, which looks good, but does not provide a detailed introduction and discussion in the related work section. I suggest the author discuss more related work.

2.The experiments should include more extreme scenarios, such as further reducing the beta value (0.1, 0.01).

3.The paper mentions using more computational resources — this is understandable, but it would be helpful to include some quantitative data on that.

Minor:

Figure 1 is too small. It seems that there is more space on page 9. The author should enlarge Figure 1.

---

> ### Author Rebuttal · Authors · 2025-07-31
>
> ### AQ1 regarding the related work：
>
> Thank you for your suggestion. We will provide a more detailed introduction to the related work in the revised version.
>
>
>
> ### AQ2:
>
> Thank you for your suggestion. Adding comparisons regarding extremely heterogeneous data will further highlight the advantages of our method. The following table results show that our method still achieved the best outcome compared to the baseline.
>
> | Model     | Setting | FedAvg      | FedProx     | FedMut      | CluSamp     | FedGen      | FedPhoenix  |
> |-----------|---------|-------------|-------------|-------------|-------------|-------------|-------------|
> | ResNet-18 | 0.01   | 29.33 ± 0.30 | 27.88 ± 0.41 | 28.20 ± 0.33 | 28.53 ± 0.26 | 29.46 ± 0.04 | 35.80 ± 0.93 |
> |  | 0.1    | 45.48 ± 0.03 | 45.41 ± 0.16 | 47.55 ± 0.14 | 45.56 ± 0.36 | 45.29 ± 7.33 | 55.80 ± 0.11 |
> | VGG16     | 0.01   | 32.24 ± 0.16 | 31.79 ± 0.37 | 32.09 ± 0.16 | 30.74 ± 0.57 | 32.03 ± 1.89 | 33.10 ± 0.41 |
> |     | 0.1    | 47.75 ± 0.11 | 48.26 ± 0.18 | 49.59 ± 0.07 | 47.72 ± 0.05 | 49.30 ± 0.29 | 52.86 ± 0.14 |
> | MobNet-V1 | 0.01   | 19.87 ± 0.31 | 19.88 ± 0.11 | 21.48 ± 0.15 | 18.31 ± 0.09 | 21.01 ± 0.14 | 27.56 ± 0.57 |
> | | 0.1    | 36.59 ± 0.18 | 36.86 ± 0.03 | 39.55 ± 0.15 | 35.48 ± 0.06 | 37.93 ± 0.13 | 50.36 ± 0.11 |
>
>
>
> ### Answer to Q3 (Analysis of Computational Overhead):
> Sorry for the confusion. We conduct a quantitative analysis of the FLOPs (Floating-Point Operations) for each round of the Reset operation.
>
> #### Notations
> -	$M$ denotes the total number of convolutional layer parameters;
> -	$K$ denotes the number of activated clients, $|S_l|$ denotes the total number of convolutional layers;
> -	$\theta \in (0,1)$ indicates the perturbation ratio, where $\lfloor \theta n_i \rfloor$ kernels are reset per layer (with $n_i$ kernels in layer $l_i$)
> -	For each layer $l_i$, $\mu_i = \mathbb{E}[W_{l_i}]$ and $\sigma_i^2 = \text{Var}(W_{l_i})$ are computed from the current layer parameters;
> -	The reset kernels are sampled from $\Delta_{l_i} \sim \mathcal{N}(\mu_i, \sigma_i^2)$.
>
> #### FLOPs estimation:
> - Computing mean $\mu_i$ requires ≈2 FLOPs per parameter (sum and divide).
> - Calculating variance $\sigma_i^2$ requires ≈4 FLOPs per parameter (subtractions, squarings, sum, and divide).
> - Sampling from $\mathcal{N}(\mu_i, \sigma_i^2)$ requires ≈10 FLOPs per parameter (including random number generation and transformation, e.g., via Box-Muller).
> - Minor operations like random kernel selection (e.g., generating masks or indices) are $O(|S_l| \times \max(n_i))$ and negligible (<1% of total overhead).
>
> #### Computation overhead of FedPhoneix
> In each round, FedPhoenix requires $2M$ FLOPs to compute the mean and $4M$ FLOPs for sampling in the cloud server.
> For the generation of each local model, FedPhoenix reset $\theta M$ parameters, and each parameter requires 10 FLOPs. Therefore, FedPhoenix requires $K \times 10 \theta M$ FLOPs for local model generation.
> In addition, the model aggregation requires $KM$ FLOPs.
> Overall, in each round, FedPhoenix requires $(6 + (10 \theta+1)K)M$ FLOPs. Note that the computation overhead of the reset operation is similar to that of the aggregation operation, and all the reset operations are performed in the cloud server. Due to the powerful computing resources of cloud servers, the additional computation overhead of the reset operation is tolerable.
>
> Additionally, we analyze the computational overhead of the baselines.
> For **FedProx**, assume that $P$ is the number of local update steps per client; its additional computation overhead is $3 P \times K \times M$ FLOPs.
> For **FedMut**, the additional computation overhead is $M(1 + 2K)$ FLOPs.
> For **ClusteredSampling**, its additional computation overhead is $3 K N M$ FLOPs, where $N$ is the total number of clients.
> Therefore, the additional computation overhead of FedPhoneix is similar to the baselines.
>
> ### AQ4：
>
> Thank you for your suggestion. We will adjust the page layout in the revised version.

---

> ### Comment · Reviewer_NLhY · 2025-08-03
> **Thanks for your rebuttal**
>
> I just carefully reviewed your responses to all the reviewers’ comments. My concerns have been largely addressed, and it seems the other reviewers’ issues have also been resolved. Please make sure the final version fills the full 9 pages. I will raise my score to 5.

---

> > ### Author Response · Authors · 2025-08-04
> >
> > Thank you for your in-depth review of our work and for carefully considering the other reviewers' comments. We will incorporate your suggestions to optimize the page layout in the final version.

---

### Official Review · Reviewer_iGvs · 2025-06-26

**Clarity:** 2
**Significance:** 2
**Originality:** 3
**Rating:** 4
**Confidence:** 4

**Summary:**

This work proposes a new FL framework called PedPhoenix to address the issue that the global model in FL
can be easily stuck in local optima due to its average aggregation strategy. It considers a stochastic reset
strategy for parameters in the global model at each round to learn generalized features more effectively.

**Questions:**

1. In the proposed dynamic stabilization strategy,
the parameter reset depends on the location of convolution layers because the criterion depends on i \times r_s
where i is the index of the convolution layer. This implies that the convolution layer with small i has less chance
to reset than the convolution layer with large i. As the communication round r increases, this becomes more severe.
Is there any reason why the criterion i \times r_s depends on the index(location in the model) of the layers?
Is it related with the fact that different convolutional layers learn different features (low-level, mid-level, high-level)
depending on their locations in the model?

2. For reset, the mean and variance of the current layer parameters are used in sampling, but the kernel contains
important feature information and each parameter contains different information. By replacing sampled values
the relative importance of each parameter seems to be disregarded. From this perspective,
is it a good strategy to replace all the parameter values by the sampled values? What happens if we select a portion
of parameters with, e.g., large values and reset them by the sampled values? The parameters with very small values
can be disregarded even in the reset because their relative importance is not significant.

3. When some features are not general but specific to some clients, its training is not conducted frequently due to
random selection of active clients and furthermore even when its training is conducted, by aggregating the weights for an
updated global model its impact becomes less significant. This seems to be true when a large number of clients involve
in FL, which is a usual setting in FL.
Even though nice experimental results in Fig. 2 and Fig. 3 are provided, the above issue needs more detailed discussion.

4. It seems that a smaller value of theta (reset proportion) is used for a more heterogeneous case (beta = 0.3, nonIID)
in section 6.1. It seems that a more heterogeneous case needs more reset portion. Is there a good reason for it?

5. In algorithm 2, the notations of the reset proportion are different in the Reset function (mu_0) and in line 7 (theta).
In the context both notations are also used to denote the reset proportion. It is better to use the same notation for a parameter.

6. In ablation study, a comparison study with Kaiming-Normal is conducted. It would be helpful if more comparison studies
with other distributions are provided because reset is the key idea and it would be interesting if the use of other
distributions for reset sampling can affect the results.

7.For non IID case, when local datasets have different features, most of which are not shared, fitting to local
datasets would rather improve the accuracy because of collaboration with which each client can learn the features that
cannot be trained with its local dataset. For instance, the input space is partitioned and each local dataset
is generated from each subset in the partition. Then, does the proposed algorithm still perform well enough?

**Ethical Concerns:**

["NO or VERY MINOR ethics concerns only"]

**Final Justification:**

I have carefully reviewed the responses. My concerns on the reset strategy and sampling distribution are well addressed and the other concerns have been also addressed. So, I am happy to raise my score to 4.

**Limitations:**

The proposed algorithm focuses on FL with convolutional neural networks and hence can be applied to limited
scenarios. A generalization of the proposed approach to other model architectures needs to be discussed.
Model homogeneity is inevitable in the proposed model, which is another limitation.

**Paper Formatting Concerns:**

No paper formatting concerns are found.

**Quality:**

3

**Strengths And Weaknesses:**

The key idea of this work is to learn more generalized features than client-specific features by stochastically
resetting a few parameters in the global model during communication rounds.
In the proposed approach, copies (with reset) of the global model are downloaded to active clients, meaning that
model homogeneity is crucial. This is a limitation. Moreover, since the global model is usually large-scaled,
downloading and uploading the copies usually require high cost.

---

> ### Author Rebuttal · Authors · 2025-07-31
>
> ### Answer to Q1
> Sorry for the confusion. Since the reasoning of deep layers in neural networks is based on the output from shallow layers, we prioritize reducing the reset rate of shallow layers to ensure training stability.
> From the perspective of feature levels, as established by seminal works such as **Zeiler & Fergus (2014, "Visualizing and Understanding Convolutional Networks")**, shallow layers learn general-purpose, low-level features (e.g., edges, textures) that stabilize quickly, and high-level features are extracted from these low-level features. Therefore, we first reduce the reset rate of shallow layers to stabilize the learned low-level features and facilitate the learning of high-level features based on stable low-level features.
>
> ### Answer to Q2
> Thanks for your insightful suggestion.
>
> Our primary motivation for resetting the *entire* kernel is to treat it as a single feature extraction unit. By completely re-initializing it, we aim to force the model to explore entirely new feature patterns, thereby enhancing generalization and feature diversity, which is the core goal of FedPhoenix.
>
> We conducted new ablation studies where we selectively reset only a portion of the parameters within a kernel based on their magnitude (top 1/3 and top 2/3). We conducted these experiments by keeping the kernel reset ratio ($\theta$) constant and by increasing it to maintain a constant total number of reset parameters. The experimental results are as follows:
>
> | Setting | Top1/3       | Top2/3       | Top1/3 Fixed θ | Top2/3 Fixed θ | FedPhoenix   |
> | ------- | ------------ | ------------ | -------------- | -------------- | ------------ |
> | 0.3     | 77.2 ± 1.87  | 78.47 ± 1.32 | 76.86 ± 0.98   | 79.55 ± 0.95   | 80.12 ± 0.54 |
> | 0.6     | 80.68 ± 0.17 | 80.9 ± 1.06  | 80.35 ± 0.05   | 81.76 ± 0.78   | 82.28 ± 0.82 |
> | IID     | 83.01 ± 0.01 | 84.78 ± 0.02 | 84.26 ± 0.01   | 84.63 ± 0.01   | 84.85 ± 0.07 |
>
>
> The results show that compared with the new plan, the original plan performs better. This is mainly because partial resets disrupt the learned structure of a kernel, potentially creating gradient conflicts during training. The holistic reset of the entire unit appears to be more effective for our purpose.
>
>
>
> ### Answer to Q3
> Sorry for the confusion. From a geometric perspective, despite limited participation of each client, learning non-generalized features will still cause the model to be stuck in a local optimum, i.e., a sharp area. Please note that due to the similarities in the characteristics of different data, in this area, the local optimum of different clients will be close but different. As the training progresses, the optimization directions of different clients will conflict, making it difficult to reach a consensus and thus getting stuck in the local optimum.
> Parameter reset can enable the model to directly jump out of the local optimal area and find a new solution. Please note that the generalization area is relatively flat. Resetting part of the model's parameters is not enough to cause the aggregated model to jump out of this area.
>
> ### Answer to Q4
> Sorry for the confusion. The optimal perturbation ratio is determined by a delicate trade-off between **promoting exploration** and **preserving the stability of learned representations**, and this balance shifts with the degree of data heterogeneity.
> Increased data heterogeneity will lead to greater differences in local model optimization directions, making it challenging to reach a consensus on the optimal direction during training.
> In these scenarios, resetting more parameters will seriously increase the time overhead of FL.
>
>
> ### Answer to Q5
> Sorry for the confusion. We will unify the notations in the final version.
>
>
> ### Answer to Q6
> Thanks for your suggestion. We conduct a comprehensive ablation study to investigate the impact of various distributions on the performance of our proposed method. We expanded our experiments to include several widely recognized and statistically diverse distributions:
>
> - **Xavier/Glorot (Normal & Uniform):** To validate the effectiveness of our reset mechanism with another mainstream initialization standard.
> - **Truncated Normal:** To evaluate a more conservative reset strategy that enhances stability by limiting extreme sample values.
> - **Orthogonal:** To test whether introducing "structured" randomness, which is known to preserve gradient norms, offers advantages over purely random sampling.
> - **Laplace:** To explore the effect of a distribution with different statistical properties (more peaked and promoting sparsity).
>
> For a fair comparison, all these distributions were implemented using the "adaptive" strategy, i.e., they were scaled using the mean and variance of the current layer's weights, consistent with our proposed `Adp_Normal` method. The experiments were conducted on the CIFAR-10 dataset and the ResNet-18 model.
>
> | Non-IID Level |  Adp_Normal  | Adp_Uniform  | Kaiming_Normal | Kaiming_Uniform | Truncated_Normal | Xavier_Normal | Xavier_Uniform | Orthogonal |   Laplace    |
> | :------------ | :----------: | :----------: | :------------: | :-------------: | :--------------: | :-----------: | :------------: | :--------------: | :----------: |
> | **0.3**       | 80.12 ± 0.54 | 80.02 ± 0.41 |  79.34 ± 0.64  |  77.58 ± 2.34   |   79.42 ± 1.45   | 79.47 ± 0.59  |  78.87 ± 2.01  | 80.00 ± 0.16 | 78.46 ± 0.34 |
> | **0.6**       | 82.28 ± 0.82 | 81.86 ± 0.24 |  79.97 ± 0.17  |  80.34 ± 0.41   |   81.70 ± 0.09   | 80.60 ± 0.21  |  81.03 ± 0.60  | 82.68 ± 0.11 | 81.74 ± 0.07 |
> | **IID**       | 85.06 ± 0.03 | 85.25 ± 0.04 |  84.70 ± 0.01  |  84.52 ± 0.03   |   85.08 ± 0.02   | 85.03 ± 0.02  |  84.75 ± 0.01  | 85.42 ± 0.03 | 85.00 ± 0.02 |
>
> Based on these results, we can find that the core strength of our method lies in the adaptive reset strategy itself, rather than the specific choice of distribution. The consistently high performance across diverse distributions (Normal, Uniform, Orthogonal, etc.) demonstrates the mechanism's robustness and low sensitivity. While the Orthogonal distribution shows a slight advantage in some cases, our proposed Adp_Normal is highly competitive across all scenarios. This confirms that adapting the perturbation to the layer's current statistics is the key factor for success, making our approach broadly effective.
>
>
>
> ### Answer to Q7
> Thanks for your suggestion. We conducted a new experiment on the **Office-31 dataset**, a standard benchmark for cross-domain scenarios, where the feature space of each client is quite different. We partitioned the dataset by its three distinct domains (**Amazon, DSLR, Webcam**), assigning each domain to a separate client. The experimental results are as follows:
>
> | Dataset   | FedAvg       | FedProx      | FedMut       | CluSamp      | FedGen       | FedPhoenix   |
> | --------- | ------------ | ------------ | ------------ | ------------ | ------------ | ------------ |
> | Office_31 | 47.84 ± 1.78 | 48.12 ± 1.68 | 43.99 ± 1.59 | 46.42 ± 1.83 | 49.13 ± 2.26 | 54.17 ± 0.96 |
>
> As shown in the table, our method achieves a **significant performance gain** over the baselines. This demonstrates that our approach not only performs well but also excels in this challenging, feature-skewed environment. By mitigating client drift and encouraging the exploration of a shared feature space, our method facilitates more effective collaboration, leading to a more robust and generalized global model.

---

> > ### Comment · Reviewer_iGvs · 2025-08-04
> >
> > I have carefully reviewed the responses. Since a number of concerns have been well addressed, I am happy to raise my score to 4.

---

> > > ### Author Response · Authors · 2025-08-04
> > >
> > > Thank you for your thoroughness and dedication. It's evident that you have deeply understood the essence of our article, and your review comments will provide invaluable inspiration for our future work.

---

### Official Review · Reviewer_QPnF · 2025-07-02

**Clarity:** 3
**Significance:** 3
**Originality:** 3
**Rating:** 5
**Confidence:** 4

**Summary:**

This paper presents a novel federated learning algorithm, named FedPhoneix, which randomly resets partial parameters to guide the FL to learn more generalized features rather than overfitting features.

**Questions:**

Please refer to the weaknesses

**Ethical Concerns:**

["NO or VERY MINOR ethics concerns only"]

**Final Justification:**

The authors have resolved the computational overhead issues I raised, and I am satisfied with their clarifications. I am willing to increase my score.

**Limitations:**

Yes

**Quality:**

3

**Strengths And Weaknesses:**

## Strengths:

* The idea of the paper is interesting and novel. The method design fits the federated learning framework rather than simply migrating existing optimization methods.

* I appreciate the explanation and pre-study in the motivation section of the paper, which exhibits the reasonability of the proposed method.

* The author provides convergence proofs, source code, and related documentation, further improving the soundness of the proposed method.

* The experiments of the paper are comprehensive, which evaluate various factors, especially exploring the adaptability with existing overfitting solutions (e.g., Dropout) and visualizing the model generalization in the appendix. The experimental results show the effectiveness of the proposed method.

## Weakness

* The author should analyze the communication and computational overhead of the method and the baselines. The author only mentions them in the appendix. In my opinion, the author should analyze them more detailed, especially quantitative comparison with baselines. In addition, it is unreasonable to put the discussion of communication and computational overhead in the limitations section.

* The author needs to modify the layout of the paper. For example, Fig 3 covers line 153 and the figure of the framework (Fig 4) and Table 2 is too small. It seems that there is still some space in the paper. The author should adjust the size of tables and figures to make full use of the space.

* The expression of some parts needs to improve. Although the overall writing of the paper is good, the author still needs to revise the expression of some sections to make it easier for readers to understand. For example, Table 1 should appear in the paragraph in which it is mentioned, and it would be better to explain Table 1 in more detail. In addition, in the discussion section at appendix, limitations, communication overhead, computation overhead, and future work should be discussed separately

---

> ### Author Rebuttal · Authors · 2025-07-31
>
> ### AQ1:Analysis of Server-Side Computational Overhead of the Parameter Reset Mechanism
>
> Sorry for the confusion. We conducted a quantitative analysis of the FLOPs (Floating-Point Operations) for each round of the Reset operation.
>
> #### Notations
> -	$M$ denotes the total number of convolutional layer parameters;
> -	$K$ denotes the number of activated clients, $|S_l|$ denotes the total number of convolutional layers;
> -	$\theta \in (0,1)$ indicates the perturbation ratio, where $\lfloor \theta n_i \rfloor$ kernels are reset per layer (with $n_i$ kernels in layer $l_i$)
> -	For each layer $l_i$, $\mu_i = \mathbb{E}[W_{l_i}]$ and $\sigma_i^2 = \text{Var}(W_{l_i})$ are computed from the current layer parameters;
> -	Reset kernels are sampled from $\Delta_{l_i} \sim \mathcal{N}(\mu_i, \sigma_i^2)$.
>
> #### FLOPs estimation:
> - Computing mean $\mu_i$ requires ≈2 FLOPs per parameter (sum and divide).
> - Computing variance $\sigma_i^2$ requires ≈4 FLOPs per parameter (subtractions, squarings, sum, and divide).
> - Sampling from $\mathcal{N}(\mu_i, \sigma_i^2)$ requires ≈10 FLOPs per parameter (including random number generation and transformation, e.g., via Box-Muller).
> - Minor operations like random kernel selection (e.g., generating masks or indices) are $O(|S_l| \times \max(n_i))$ and negligible (<1% of total overhead).
>
> #### Computation overhead of FedPhoneix
> In each round, FedPhoenix requires $2M$ FLOPs to compute the mean and $4M$ FLOPs for sampling in the cloud server.
> For the generation of each local model, FedPhoenix reset $\theta M$ parameters, and each parameter requires 10 FLOPs. Therefore, FedPhoenix requires $K \times 10 \theta M$ FLOPs for local model generation.
> In addition, the model aggregation requires $KM$ FLOPs.
> Overall, in each round, FedPhoenix requires $(6 + (10 \theta+1)K)M$ FLOPs. Note that the computation overhead of the reset operation is similar to that of the aggregation operation, and all the reset operations are performed in the cloud server. Due to the powerful computing resources of cloud servers, the additional computation overhead of the reset operation is tolerable.
>
> Additionally, we analyze the computational overhead of the baselines.
> For **FedProx**, assume that $P$ is the number of local update steps per client; its additional computation overhead is $ 3P \times K \times M$ FLOPs.
> For **FedMut**, the additional computation overhead is $M(1 + 2K)$ FLOPs.
> For **ClusteredSampling**, the additional computation overhead is $3 K N M$ $FLOPs, where $N$ is the total number of clients.
> Therefore, the additional computation overhead of FedPhoneix is similar to the baselines.
>
> ### AQ: About writing optimization
>
> Thank you for your rich writing experience. We will make adjustments to Figure 3 and Figure 4 in the revised version. And further optimize the presentation of the core concepts (such as Table 1). In the discussion section, we will refine the framework to make writing more coherent and organized.

---

> > ### Comment · Reviewer_QPnF · 2025-08-04
> > **Thanks for your rebuttal**
> >
> > After reading the authors’ rebuttal and their responses to other reviewers, I find that my main concerns, particularly those related to the computational overhead, have been addressed. I decide to increase my original score.

---

> > > ### Author Response · Authors · 2025-08-05
> > >
> > > Thank you for your response. We are glad that our rebuttal addressed your concerns about the computational overhead.
> > > We are very grateful for your decision to increase your score; your support means a lot to us.

---

### Official Review · Reviewer_WVfY · 2025-07-03

**Clarity:** 3
**Significance:** 2
**Originality:** 3
**Rating:** 4
**Confidence:** 2

**Summary:**

This paper introduces FedPhoenix, an FL framework designed to enhance model generalization and inference performance in non-IID (non-independent and identically distributed) data scenarios. The core concept of FedPhoenix is a server-side dynamic parameter reset mechanism. This mechanism stochastically resets partial parameters of the global model before dispatching them to clients for local training, aiming to disrupt client-specific overfitting features and encourage the learning of more generalized features across the federated network. The authors contribute a novel FL framework with an adaptive parameter reset strategy, a dynamic stabilization strategy, theoretical analysis for convergence, and comprehensive empirical evaluations demonstrating significant accuracy improvements over state-of-the-art FL methods on various datasets and model architectures.

**Questions:**

**-** Could you elaborate on the practical implications and potential adaptations of FedPhoenix for Transformer architectures and NLP tasks?

Actionable Guidance: Please discuss how the parameter reset mechanism would apply to specific components of Transformers. A conceptual outline of how the destruction and restoration of features would manifest in this domain, and any anticipated challenges or necessary modifications, would be highly valuable.


**-** Please provide a more detailed analysis of the server-side computational overhead introduced by the parameter reset mechanism.
Actionable Guidance: Quantify the computational cost (FLOPs or actual wall-clock time) of the Reset function and the overall server-side processing per round. Present this data as a function of model size (number of parameters) and the number of activated clients K to better assess scalability.


**-** What is the rationale behind exclusively applying the parameter reset to convolutional layers and not to fully connected layers?
Actionable Guidance: Please provide a justification for this design choice.


**-** Could you provide a more precise characterization of the privacy-preserving aspect of FedPhoenix beyond increasing uncertainty and diversity?
Actionable Guidance: Clarify whether the current mechanism offers any formal privacy guarantees against specific attack models like membership inference or model inversion.

**Ethical Concerns:**

["NO or VERY MINOR ethics concerns only"]

**Final Justification:**

I keep my initial score, which is already high enough. I believe the camera-ready should have sufficient time to address the issues.

**Limitations:**

In Appendix C.2, the authors acknowledge several limitations. I would argue that some of them should move to the main text.

**-** The ablation studies in Figure 7 show that the performance of FedPhoenix is sensitive to the reset rounds and reset rates hyperparameters. While the paper states that all variants eventually converge, achieving optimal performance appears to depend on careful tuning. In real-world FL settings, where validation data might be limited or difficult to obtain, the sensitivity to these parameters could hint at a practical challenge for deployment.

**-** The core concept relies on "destroying" features to encourage generalization. While effective for CNNs, the interpretation, and effectiveness of this "destruction" might vary significantly for different types of neural network architectures (RNN for sequential data, GNN for graph data) because of different structural implications.

**Quality:**

2

**Strengths And Weaknesses:**

Strengths

**+** The central idea of dynamic parameter reset on the server side is a genuinely innovative approach to combat overfitting in non-IID federated learning.

**+**  The paper clearly articulates the problem of models overfitting to local client data and provides compelling empirical evidence to demonstrate that existing FedAvg models indeed rely on a narrow set of specific features.

**+** This design avoids the common pitfalls of client-side modifications, additional communication overhead, or the need for public proxy datasets, which often limit the practicality and deployability of other FL debiasing or generalization methods.

Areas for improvement

**-**  Given the increasing prominence of Transformers in FL, empirical validation is exclusively confined to vision tasks and CNNs; this significantly limits the scope of generalization implied.

**-** The authors state that the server-side computational overhead for generating multiple model copies with reset parameters is comparable to FedAvg. However, the detailed quantification of this overhead is not fully convincing. For very large models or a significantly higher number of active clients (K), this overhead could become a practical bottleneck.

**-** Fully connected layers are often critical for final classification and can also be prone to overfitting. The rationale for this specific focus, and whether extending the parameter reset mechanism to these layers would offer further benefits or introduce new challenges, is not thoroughly explored.

**-** While introducing noise can have privacy benefits, this is a weaker claim than formal privacy guarantees (e.g., differential privacy). The paper itself suggests integrating differential privacy in future work. Clarifying the specific type of privacy enhancement offered by the current mechanism and distinguishing it from more formal privacy guarantees would provide a more accurate picture.

---

> ### Author Rebuttal · Authors · 2025-07-31
>
> ### Answer to Q1/W1 (Transformer)
> This paper primarily focuses on computer vision-based tasks. As mentioned in the Limitation section of the paper, since the different structure of Transformer and CNN, the current version of FedPhoenix cannot adapt to Transformer-based models.
>
> However, we sincerely thank the reviewer for this forward-thinking question. While much of the current research in Federated Learning has concentrated on CNNs for vision tasks—largely due to the proliferation of resource-constrained AIoT devices—we strongly agree that extending these privacy-preserving and collaborative learning paradigms to Large Language Models is a highly valuable research direction and a key focus for our future work.
>
> **Conceptual Feasibility: Applying FedPhoenix to Mitigate Transformer Drift**
>
> The core philosophy of FedPhoenix is to **prevent the model from overfitting to local data by periodically resetting a portion of its learned features. These reset units, possessing high plasticity, are then compelled to rediscover more generalizable patterns.** Furthermore, the FFNs in Transformers utilize fully-connected layers. As shown in our experimental results in AQ3, when a model has a robust feature extraction layer, resetting the fully-connected structures can also lead to performance improvements.
>
> Rather than a full parameter reset, which could excessively disrupt the highly structured knowledge within a Transformer, we can adapt the "reset" operation into a more refined **blending mechanism**. For a selected parameter `W`, the update formula would be:
>
> ```
> W_new = (1 - α) * W_old + α * W_sampled
> ```
>
> Here, `W_sampled` is a new value sampled from a normal distribution `N(μ, σ)` based on layer-wise statistics, and `α` is a blending factor (analogous to the original parameter `ε`, but controlling the intensity of the blend rather than the proportion of reset units).
>
> This approach does not cause complete "amnesia" but rather **"nudges" the parameters away from a highly specialized state**. It helps to loosen the tight coupling between specific attention patterns or word embeddings and the data distribution of a single client (e.g., a "developer" client might associate "Python" with the programming language, whereas a "zoologist" client might associate it with the reptile), encouraging the model to build robust representations that span across client domains.
>
> **Operational Guidelines: Target Modules and Reset Granularity**:
>
> - **Self-Attention Mechanism (`W_q`, `W_k`, `W_v`, `W_o` matrices)**:
>   - **Target**: The query, key, value, and output projection matrices are prime candidates, as they directly govern the learned relationships between tokens.
>   - **Granularity**: We would apply the blending operation to entire rows or columns of these matrices. Conceptually, this is equivalent to perturbing the "functional role" of specific feature dimensions within the attention mechanism.
> - **Feed-Forward Networks (FFN)**:
>   - **Target**: The two linear transformation layers within each Transformer block.
>   - **Granularity**: Resetting the incoming weights for individual output neurons.
>
> **Anticipated Challenges and a Dynamic Attenuation Strategy**
>
> - **Hyperparameter Sensitivity**: The blending factor `α` and the proportion of parameters to be perturbed are critical hyperparameters that will require careful tuning to balance knowledge retention and exploration.
> - The *dynamic stabilization strategy* would be adapted as follows: **apply stronger/more frequent perturbations to the earlier Transformer blocks during the initial communication rounds.** As training progresses, the focus of the perturbation would gradually shift to the later, more abstract blocks. This approach maintains the original design philosophy of first solidifying foundational features while preserving the plasticity of higher-level representations.
>
> We thank the reviewer again for this suggestion and will incorporate this discussion into the "Future Work" section of our paper to broaden its scope.
>
> ### Answer to Q2/W2 (Analysis of Computational Overhead):
> Sorry for the confusion. We conducted a quantitative analysis of the FLOPs (Floating-Point Operations) for each round of the Reset operation.
>
> #### Notations
> -	$M$ denotes the total number of convolutional layer parameters;
> -	$K$ denotes the number of activated clients, $|S_l|$ denotes the total number of convolutional layers;
> -	$\theta \in (0,1)$ indicates the perturbation ratio, where $\lfloor \theta n_i \rfloor$ kernels are reset per layer (with $n_i$ kernels in layer $l_i$)
> -	For each layer $l_i$, $\mu_i = \mathbb{E}[W_{l_i}]$ and $\sigma_i^2 = \text{Var}(W_{l_i})$ are computed from the current layer parameters;
> -	Reset kernels are sampled from $\Delta_{l_i} \sim \mathcal{N}(\mu_i, \sigma_i^2)$.
>
> #### FLOPs estimation:
> - Computing mean $\mu_i$ requires ≈2 FLOPs per parameter (sum and divide).
> - Calculating variance $\sigma_i^2$ requires ≈4 FLOPs per parameter (subtractions, squarings, sum, and divide).
> - Sampling from $\mathcal{N}(\mu_i, \sigma_i^2)$ requires ≈10 FLOPs per parameter (including random number generation and transformation, e.g., via Box-Muller).
> - Minor operations such as random kernel selection (e.g., generating masks or indices) have a time complexity of $O(|S_l| \times \max(n_i))$, which is negligible (<1% of total overhead) in practice.
>
> #### Computation overhead of FedPhoneix
> In each round, FedPhoenix requires $2M$ FLOPs to compute the mean and $4M$ FLOPs for sampling in the cloud server.
> For the generation of each local model, FedPhoenix reset $\theta M$ parameters, and each parameter requires 10 FLOPs. Therefore, FedPhoenix requires $K \times 10 \theta M$ FLOPs for local model generation.
> In addition, the model aggregation requires $KM$ FLOPs.
> Overall, in each round, FedPhoenix requires $(6 + (10 \theta+1)K)M$ FLOPs. Note that the computation overhead of the reset operation is similar to that of the aggregation operation, and all the reset operations are performed in the cloud server. Due to the powerful computing resources of cloud servers, the additional computation overhead of the reset operation is tolerable.
>
> Additionally, we analyze the computational overhead of the baselines.
> For **FedProx**, assume that $P$ is the number of local update steps per client; its additional computation overhead is $3 P \times K \times M$ FLOPs.
> For **FedMut**, the additional computation overhead is $M(1 + 2K)$ FLOPs.
> For **ClusteredSampling**, the additional computation overhead is $3 K N M$ $FLOPs, where $N$ is the total number of clients.
> Therefore, the additional computation overhead of FedPhoneix is similar to the baselines.
>
>
> ### Answer to Q3/W3 (Reset for full-connected layers):
>
> Sorry for the confusion.
> As mentioned in the motivation of our paper, we aim to adopt parameter reset to guide the model to learn more generalized features.
> Typically, since CNN-based models employ convolutional layers to construct feature extractors, we prefer to reset the parameters of these convolutional layers.
> Note that, based on the inefficient feature extractor, it is difficult to improve the model performance by simply resetting the fully connected layer (i.e., the classifier).
> We conducted experiments on Cifar-10 using the VGG model to evaluate the effectiveness of resetting fully connected (FC) layers.
> The experimental results are as follows:
> | Model | Setting | Conv_Fc      | Only Conv    | Only Fc      | FedAvg       |
> |-------|---------|--------------|--------------|--------------|--------------|
> | VGG   | 0.3     | 83.32 ± 0.18 | 81.46 ± 0.70 | 76.81 ± 0.27 | 75.92 ± 1.16 |
> | VGG   | 0.6     | 83.84 ± 0.58 | 83.24 ± 0.35 | 78.46 ± 0.77 | 79.17 ± 0.06 |
> | VGG   | IID     | 87.18 ± 0.04 | 87.17 ± 0    | 79 ± 0.02    | 80.85 ± 0.01 |
>
> We can observe that only resetting FC cannot effectively improve the inference accuracy, but resetting both the convolutional layers and the fully connected layers can achieve the best performance.
>
> ### Answer to Q4/W4 (Pravicy):
> Thanks for your suggestion. Note that the **primary objective of FedPhoenix is to enhance the generalization and robustness of the model**, which cannot provide the formal (ε, δ)-guarantee in DP.
> Since FedPhoenix resets part of the parameters, attackers must make more effort to apply model inversion attacks.
> In addition, since the parameter reset is based solely on the aggregated global model, FedPhoneix supports secure aggregation. Furthermore, enabling us to handle the model reset operation at the client end will result in better protection against malicious server model reverse attacks. This is because each activated client will have a different training starting point. From the analysis of the computational cost of AQ2, it can also be seen that the cost of our Reset operation is relatively small, and each client can fully bear the cost of resetting their model.

---

### Note · Authors · 2025-08-14

**Dear Area Chair and Senior Area Chairs**,

We thank all four reviewers (WVfY, QPnF, iGvs, and NLhY) for their constructive feedback. We provided comprehensive responses and performed important new experiments, leading to score improvements from three reviewers (QPnF, iGvs, and NLhY). All reviewers agree that their primary concerns have been resolved.

**1. Addressed Concerns on Practicality and Computational Overhead (Reviewers WVfY & QPnF & NLhY)**

**Issue:** The initial manuscript did not include a quantitative analysis of the computational overhead.

**Our Action:** We provided a formal FLOPs analysis comparing FedPhoenix to baselines. This analysis showed that the server-side overhead of our method is similar to standard aggregation and is completely practical.

**2. Validated Methodological Rigor and Robustness (Reviewers iGvs & NLhY)**

**Issue:** Questions on core design choices (e.g., layer selection, reset granularity) and requests for testing under more extreme Non-IID and feature-skewed conditions.

**Our Action:** We conducted a series of new targeted experiments to provide strong empirical evidence.

| Experiment | Evidence |
|------------|----------|
| Extreme Non-IID (β=0.01, 0.1) | Achieving significant performance gains over baselines, directly addressing the request of NLhY |
| Feature-Skewed Data (Office-31) | Demonstrating superior performance in a challenging domain-shift scenario, addressing iGvs's concern |
| Ablation Studies | We empirically confirmed our design decisions regarding reset granularity (whole-kernel versus partial) and demonstrated robustness across both a range of sampling distributions (such as Normal, Uniform, and Orthogonal) and different types of layers. |

**3. Clarified Scope, Generalization, and Privacy (Reviewers WVfY)**

**Issue:** The scope of the research appeared limited to common models in traditional federated learning, and privacy claims needed clarification.

**Our Action:**

| Area | Solution |
|------|----------|
| Transformers | We provided a conceptual framework for adapting FedPhoenix to Transformers, outlining a "blending" mechanism and identifying target modules |
| Privacy | We clarified that our method improves privacy by making model inversion attacks more challenging, while candidly acknowledging that it does not provide formal Differential Privacy guarantees |

The review process has significantly strengthened our paper. We look forward to your favorable consideration of our paper.

---

### Decision · Program_Chairs · 2025-09-17

**Decision:**

Accept (poster)

**Comment:**

The work introduces FedPhoenix, a novel FL framework designed to mitigate overfitting in non-IID settings through a server-side dynamic parameter reset mechanism.

The primary strengths of this submission are its novelty and soundness. Reviewers consistently praised the central idea as interesting, innovative, and well-suited to the federated learning paradigm, avoiding common implementation pitfalls like increased communication overhead or the need for public proxy data. The authors provide a clear motivation for their approach, theoretical analysis for convergence, and comprehensive empirical evaluations that convincingly demonstrate significant accuracy improvements over state-of-the-art methods.

Furthermore, the authors engaged constructively during the rebuttal period, providing new experiments and clarifications that addressed the majority of reviewers' concerns, leading to score increases from three of the four reviewers. All reviewers ultimately agreed that their main issues were resolved.

Nevertheless, here are several areas for improvement that should be addressed in the final camera-ready version. Firstly, the most significant limitation noted by reviewers is the empirical scope; experiments are confined to CNNs, while validation on more modern architectures like Transformers is missing. The final paper would be substantially strengthened by including at least preliminary Transformer experimental results or by clearly positioning this as a critical avenue for future work. Secondly,  reviewers noted that the analysis of computational overhead could be more detailed and needs better visibility. Finally, the paper should include a more thorough discussion on the sensitivity to hyperparameters like reset rates and rounds, which can be included in the supplement.

In summary, this is a technically solid paper with a novel contribution and strong results. The reasons to accept decidedly outweigh the weaknesses, and incorporating the suggested revisions will further enhance the quality and impact of this work.